# Dual Caption Preference Optimization for Diffusion Models

## Abstract

Recent advancements in human preference optimization, originally developed for Large Language Models (LLMs), have shown significant potential in improving text-to-image diffusion models. These methods aim to learn the distribution of preferred samples while distinguishing them from less preferred ones. However, existing preference datasets often exhibit overlap between these distributions, leading to a *conflict distribution*. Additionally, we identified that input prompts contain irrelevant information for less preferred images, limiting the denoising network's ability to accurately predict noise in preference optimization methods, known as the *irrelevant prompt* issue. To address these challenges, we propose **Dual Caption Preference Optimization (DCPO)**, a novel approach that utilizes two distinct captions to mitigate irrelevant prompts. To tackle conflict distribution, we introduce the *Pick-Double Caption* dataset, a modified version of Pick-a-Pic v2 with separate captions for preferred and less preferred images. We further propose three different strategies for generating distinct captions: captioning, perturbation, and hybrid methods. Our experiments show that DCPO significantly improves image quality and relevance to prompts, outperforming Stable Diffusion (SD) 2.1, $SFT_{Chosen}$, Diffusion-DPO and MaPO across multiple metrics, including Pickscore, HPSv2.1, GenEval, CLIPscore, and ImageReward, fine-tuned on SD 2.1 as the backbone.

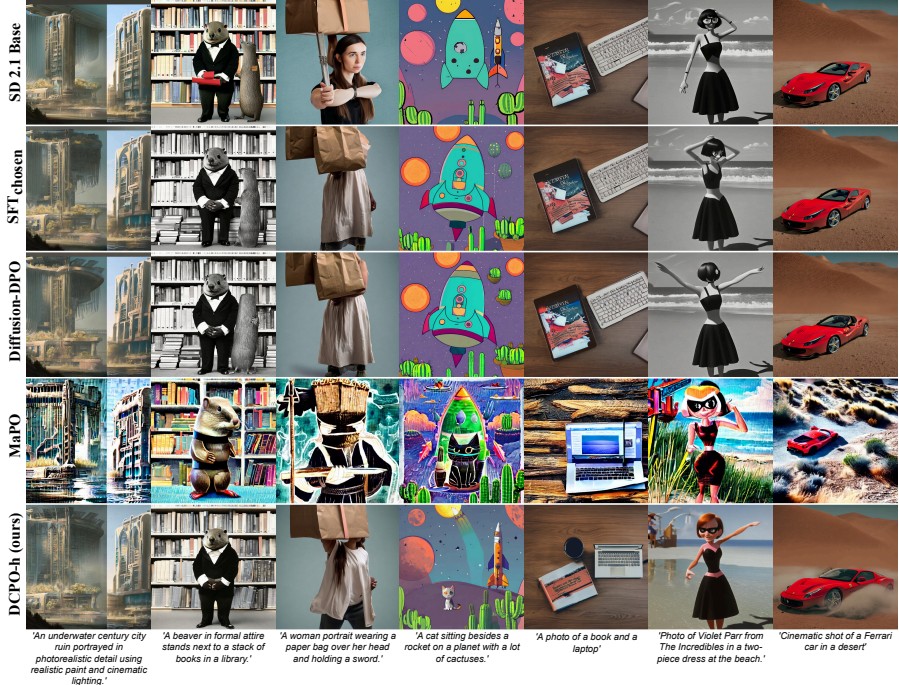

*'An underwater century city ruin portrayed in photorealistic detail using realistic paint and cinematic lighting.'*    *'A beaver in formal attire stands next to a stack of books in a library.'*    *'A woman portrait wearing a paper bag over her head and holding a sword.'*    *'A cat sitting besides a rocket on a planet with a lot of cactuses.'*    *'A photo of a book and a laptop'*    *'Photo of Violet Parr from The Incredibles in a two-piece dress at the beach.'*    *'Cinematic shot of a Ferrari car in a desert'*

Figure 1: Sample images generated by different methods on the HPSv2, Geneval, and Pickscore benchmarks. After fine-tuning SD 2.1 with $SFT_{Chosen}$, Diffusion-DPO, MaPO, and DCPO on Pick-a-Picv2 and Pick-Double Caption datasets, DCPO produces images with notably higher preference and visual appeal (See more examples in Appendix F).

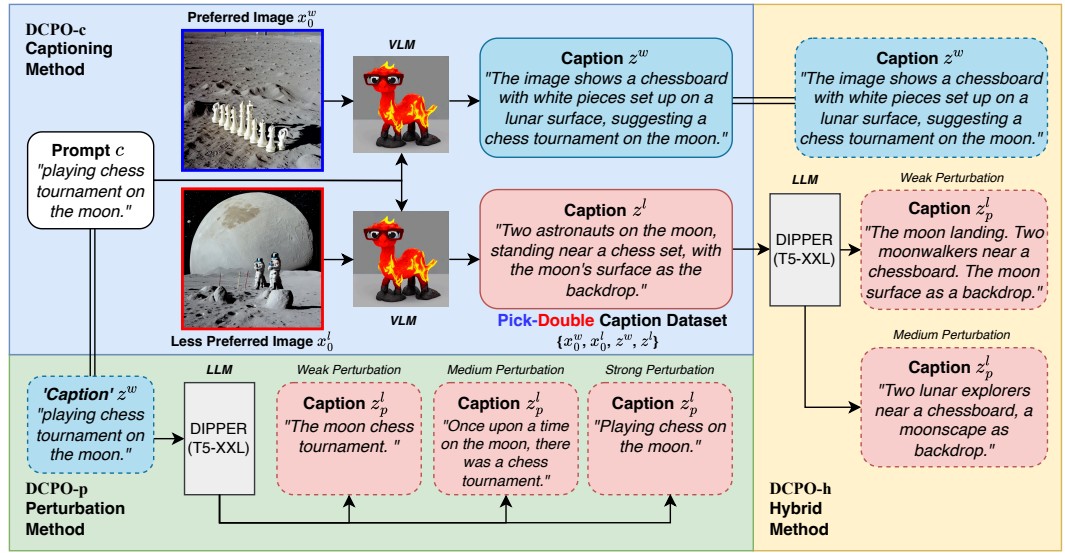

Figure 2: The overview of our 3 Dual Preference Optimization (DCPO) pipelines: DCPO-c, DCPO-p, and DCPO-h, all of which require a duo of a captioned preferred image $(x_0^w, z^w)$ and a captioned less-preferred image $(x_0^l, z^l)$. **DCPO-c (Top Left):** We use a captioning model to generate distinctive captions respectively for images $x_0^w$ and $x_0^l$ given the shared prompt $c$. **DCPO-p (Bottom Left):** We take prompt $c$ as the caption for image $x_0^w$, then we use a Large Language Model (LLM) to generate a semantically perturbed prompt $z_p^l$ given prompt $c$ as the caption for image $x_0^l$. **DCPO-h (Right):** A hybrid method where the generated caption $z^l$ is now perturbed into $z_p^l$ for image $x_0^l$. Our *Pick-Double Caption* Dataset discussed in Section 3.1 is constructed with the DCPO-c pipeline.

# 1 INTRODUCTION

Image synthesis models (Rombach et al., 2022; Esser et al., 2024) have achieved remarkable advancements in generating photo-realistic and high-quality images. Text-conditioned diffusion (Song et al., 2020a) models have led this progress due to their strong generalization abilities and proficiency in modeling high-dimensional data distributions. As a result, they have found wide range of applications in image editing (Brooks et al., 2023), video generation (Wu et al., 2023a) and robotics (Carvalho et al., 2023). Consequently, efforts have focused on aligning them with human preferences, targeting specific attributes like safety (Liu et al., 2024b), style (Everaert et al., 2023), and personalization (Ruiz et al., 2023), thereby improving their usability and adaptability.

Similar to the alignment process of Large Language Models (LLMs), aligning diffusion models involves two main steps: **1.** Pre-training and **2.** Supervised Fine-Tuning (SFT). Recent fine-tuning based methods have been introduced to optimize diffusion models according to human preferences by leveraging Reinforcement Learning with Human Feedback (RLHF) (Ouyang et al., 2022), the aim of which is to maximize an explicit reward. However, challenges such as fine-tuning a separate reward model and reward hacking have led to the adoption of Direct Preference Optimization (DPO) (Rafailov et al., 2024) techniques like Diffusion-DPO (Wallace et al., 2024). Intuitively, Diffusion-DPO involves maximizing the difference between a preferred image and a less preferred image for a given prompt.

Although DPO-based methods are effective in comparison to SFT-based approaches, applying direct optimization in multi-modal settings presents certain challenges. Current preference optimization datasets consist of a preferred ($x^w$) and a less preferred ($x^l$) image for a given prompt ($c$). Ideally, $x^w$ should show a higher correlation with $c$ compared to $x^l$. However, we find that in current datasets, both the images share the same distribution for the given prompt $c$, which we refer to as *conflict distribution* in the data. Additionally, irrelative information in $c$ restricts the U-Net's ability to predict noises from $x^l$ in the diffusion reverse process, which we refer to as *irrelevant prompts*. This entails that there is a lack of sufficient distinguishing features between the two pairs ($x^w$, $c$), ($x^l$, $c$), thereby increasing the complexity of the optimization process.

Overall, we identify two key challenges: **1.** *conflict distribution* in data, which contradicts the core idea of direct optimization, and **2.** *irrelevant prompts* for less preferred images, which can hinder the learning process of the diffusion model during optimization.

To address the aforementioned bottleneck, we propose **DCPO: Dual Caption Preference Optimization**, a novel preference optimization technique designed to align diffusion models by utilizing two distinct captions corresponding to the preferred and less preferred image. DCPO broadly consists of two steps - a text generation framework that develops better aligned captions and a novel objective function that utilizes these captions as part of the training process.

The text generation framework seeks to alleviate the *conflict distribution* present in existing datasets. We hypothesize that $c$ does not serve as the optimal signal for optimization because they do not convey the reasons why an image is preferred or dis-preferred; based on the above, we devise the following techniques to generate better aligned captions. The *first* method involves using a captioning model $Q_\phi(z^i|x^i, c)$; which generates a new prompt $z^i$ based on an image $x^i$ and the original prompt $c$, where $i \in (w, l)$. The *second* method introduces perturbation techniques $f$, such that $c = z^w, z^l = f(c)$; i.e. generating $z^l$, to represent the less preferred image, considering the original prompt $c$ as the prompt aligned with the preferred image. We investigate multiple semantic variants of $f$, where each variant differs in the degree of perturbation applied to the original caption $c$. Finally, we also explore a hybrid combination of the above methods, where we combine the strong prior of the captioning model and the efficient nature of the perturbation method. All the above methods are designed to generate captions that effectively discriminate between the preferred and less preferred images.

We introduce a novel objective function that allows DCPO to incorporate $z^w$ and $z^l$ into its optimization process. Specifically, during optimization, the policy model $p_\theta$ increases the likelihood of the preferred image $x^w$ conditioned on the prompt $z^w$, while simultaneously decreasing the likelihood of the less preferred image $x^l$ conditioned on the prompt $z^l$. The results in Tables 1 and 2 demonstrate that DCPO consistently outperforms other methods, with notable improvements of +0.21 in Pickscore, +0.45 in HPSv2.1, +1.8 in normalized ImageReward, +0.15 in CLIPscore, and +3% in GenEval. Additionally, DCPO achieved 58% in general preference and 66% in visual appeal compared to Diffusion-DPO on the PartiPrompts dataset, as evaluated by GPT-4o (see Figure 8).

In summary, our contributions are as follows :

- **Double Caption Generation**: We introduce the Captioning and Perturbation methods to address the conflict distribution issue, as illustrated in Figure 3. In the Captioning method, we employ state-of-the-art models like LLaVA (Liu et al., 2024a) and Emu2 (Sun et al., 2024) to generate a caption $z$ based on the image $x$ and prompt $c$. Additionally, we use DIPPER (Krishna et al., 2024), a paraphrase generation model built by fine-tuning the T5-XXL model to create three levels of perturbation from the prompt $c$.

- **Dual Caption Preference Optimization (DCPO):** We propose DCPO, a modified version of Diffusion-DPO, that leverages the U-Net encoder embedding space for preference optimization. This method enhances diffusion models by aligning them more closely with human preferences, using two distinct captions for the preferred and less preferred images during optimization.

- **Improved Model Performance:** We demonstrate that our approach significantly outperforms SD 2.1, SFT, Diffusion-DPO, and MaPO across metrics such as Pickscore, HPSv2.1, GenEval, CLIPscore, normalized ImageReward, and GPT-4o (Achiam et al., 2023) evaluations.

## 2 METHOD

In this section, we present the *conflict distribution* issue in preference datasets, where preferred and less-preferred images generated from the same prompt $c$ exhibit significant overlap. We also explain the *irrelevant prompt* issue found in previous direct preference optimization methods. To address these challenges, we propose **Dual Caption Preference Optimization (DCPO)**, a method that uses distinct captions for preferred and less preferred images to improve diffusion model alignment.

## 2.1 THE CHALLENGES

Generally, to optimize a Large Language Model (LLM) using preference algorithms, we need a dataset $D = \{c, y^w, y^l\}$, where $y^w$ and $y^l$ represent the preferred and less preferred responses to a given prompt $c$. Ideally, the distributions of these responses should differ significantly. Similarly, in diffusion model alignment, the distributions of preferred and less preferred images should be distinct for the same prompt $c$. However, our analysis shows a substantial overlap between these distributions, which we call *conflict distribution*, as illustrated in Figure 3.

Another issue emerges when direct preference optimizes a diffusion model. In the reverse denoising process, the U-Net model predicts noise for both preferred and less preferred images using the same prompt $c$. As prompt $c$ is more relevant to the preferred image, it becomes less effective for predicting the less preferred one, leading to reduced performance. We call this the *irrelevant prompts* problem.

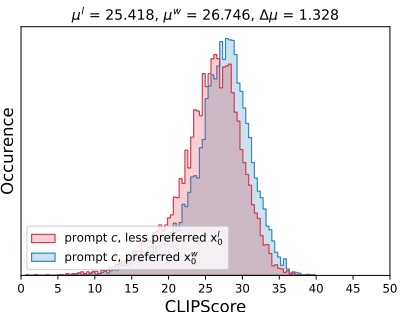

Figure 3: The *conflict distribution* issue in the Pick-a-Pic v2 dataset. $\mu^l$ and $\mu^w$ represent the average CLIPscore of preferred and less preferred images for prompt $c$, respectively. Also, $\Delta\mu$ shows the difference between the distributions.

## 2.2 DCPO: DUAL CAPTION PREFERENCE OPTIMIZATION

Motivated by the *conflict distribution* and *irrelevant prompts* issues, we propose DCPO, a new preference optimization method that optimizes diffusion models using two distinct captions. DCPO is a refined version of Diffusion-DPO designed to address these challenges.[1]

We start with a fixed dataset $D = \{c, x_0^w, x_0^l\}$, where each entry contains a prompt $c$ and a pair of images generated by a reference model $p_{ref}$. The human labels indicate a preference, with $x_0^w$ preferred over $x_0^l$. We assume the existence of a model $R_\phi(z|c, x)$, which generates a caption $z$ given a prompt $c$ and an image $x$. Using this model, we transform the dataset into $D' = \{z^w, z^l, x_0^w, x_0^l\}$, where $z^w$ and $z^l$ are captions for the preferred image $x_0^w$ and the less-preferred image $x_0^l$, respectively. Our goal is to train a new model $p_\theta$, aligned with human preferences, to generate outputs that are more desirable than those produced by the reference model.

The objective of RLHF is to maximize the reward $r(c, x_0)$ for the reverse process $p_\theta(x_{0:T}|z)$, while maintaining alignment with the original reference reverse process distribution. Building on prior work (Wallace et al., 2024), the DCPO objective is defined by direct optimization through the conditional distribution $p_\theta(x_{0:T}|z)$ as follows:

$$\mathcal{L}_{\text{DCPO}}(\theta) = -\mathbb{E}_{(x_0^w, x_0^l)\sim\mathcal{D}'} \log \sigma(\beta\mathbb{E}_{x_{1:T}^w\sim p_\theta(x_{1:T}^w|x_0^w, z^w), x_{1:T}^l\sim p_\theta(x_{1:T}^l|x_0^l, z^l)}$$
$$[\log \frac{p_\theta(x_{0:T}^w|z^w)}{p_{\text{ref}}(x_{0:T}^w|z^w)} - \log \frac{p_\theta(x_{0:T}^l|z^l)}{p_{\text{ref}}(x_{0:T}^l|z^l)}]) \tag{1}$$

However, as noted in Diffusion-DPO (Wallace et al., 2024), the sampling process $x_{1:T} \sim p(x_{1:T}|x_0)$ is inefficient and intractable. To overcome this, we follow a similar approach by applying Jensen's inequality and utilizing the convexity of the $-\log(\cdot)$ function to bring the expectation outside. By approximating the reverse process $p_\theta(x_{1:T}|x_0, z)$ with the forward process $q(x_{1:T}|x_0)$, and through algebraic manipulation and simplification, the DCPO loss can be expressed as:

$$\mathcal{L}_{\text{DCPO}}(\theta) = -\mathbb{E}_{(x_0^w, x_0^l)\sim\mathcal{D}', t\sim\mu(0,T), x_t^w\sim q(x_t^w|x_0^w), x_t^l\sim q(x_t^l|x_0^l)}$$
$$\log \sigma(-\beta T w(\lambda_t)(||\epsilon^w - \epsilon_\theta(x_t^w, z^w, t)||_2^2 - ||\epsilon^w - \epsilon_{\text{ref}}(x_t^w, z^w, t)||_2^2 \tag{2}$$
$$-(||\epsilon^l - \epsilon_\theta(x_t^l, z^l, t)||_2^2 - ||\epsilon^l - \epsilon_{\text{ref}}(x_t^l, z^l, t)||_2^2))$$

---

[1]For additional background about diffusion and preference optimization, refer to Appendix A.

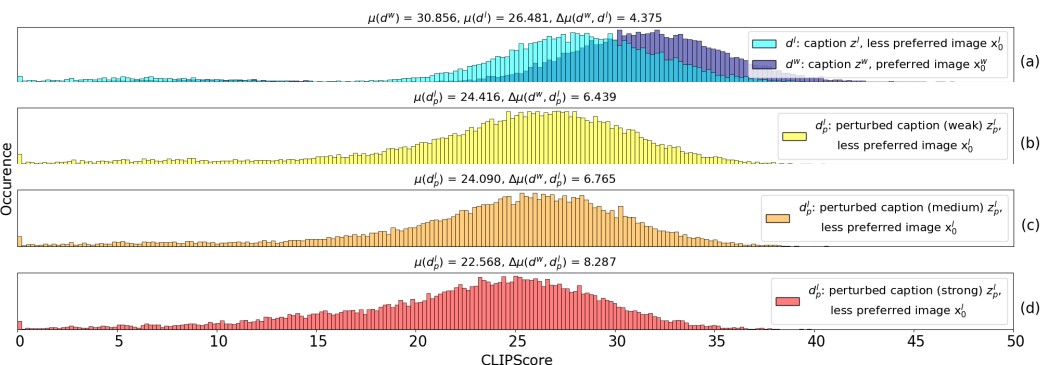

Figure 4: Effect of the perturbation method on semantic distributions in terms of CLIPScore. **(a)** shows the distributions that feature the captions $z^w$ and $z^l$ generated by the LLaVA model, while **(b)**, **(c)**, and **(d)** represent different levels of perturbation on top of the caption $z^l$. The figure demonstrates that as the level of perturbation increases, the distance between the distributions of captions $z^w$ and $z^l$ increases. For more details on the perturbation method, refer to Appendix C.

where $x_t^* = \alpha_t x_0^* + \sigma_t \epsilon^*$, and $\epsilon^* \sim \mathcal{N}(0, I)$ is a sample drawn from $q(x_t^*|x_0^*)$. $\lambda_t = \alpha_t^2/\sigma_t^2$ represents the signal-to-noise ratio, and $\omega(\lambda_t)$ is a weighting function.

To optimize a diffusion model using DCPO, a dataset $D = \{z^w, z^l, x_0^w, x_0^l\}$ is required, where captions are paired with the images. However, the current preference dataset only contains prompts $c$ and image pairs without captions. To address this, we propose three methods for generating captions $z$ and introduce a new high-quality dataset, *Pick-Double Caption*, which provides specific captions for each image, based on Pick-a-Pic v2 (Kirstain et al., 2023).

### 2.2.1 DCPO-c: CAPTIONING METHOD

In this method, the captioning model $Q_\phi(z|c, x)$ generates the caption $z$ based on the image $x$ and the original prompt $c$. As a result, we obtain a preferred caption $z^w \sim Q_\phi(z^w|c, x^w)$ for the preferred image and a less preferred caption $z^l \sim Q_\phi(z^l|c, x^l)$ for the less preferred image, as illustrated in a sample in Figure 2. Thus, based on the generated captions $z^w$ and $z^l$, we can optimize a diffusion model using the DCPO method.

In the experiment section, we evaluate the performance of DCPO-c and demonstrate that this method effectively mitigates the *conflict distribution* by creating two differentiable distributions. However, the question of how much divergence is needed between the two distributions remains. To investigate this, we propose Hypothesis 1.

**Hyphothesis 1.** *Let $d(z, x)$ represent the semantic distribution between a caption $z$ and an image $x$, with $\mu$ being the mean of the distribution $d$, and $\Delta\mu = \mu(d(z_0^w, x_0^w)) - \mu(d(z_0^l, x_0^l))$ as the difference between the two distributions. Increasing $\Delta\mu$ between the preferred and less-preferred image distributions in a preference dataset beyond a threshold $t$ (i.e., $\Delta\mu > t$), can improve the performance of the model $p_\theta$.*

Our hypothesis suggests that increasing the distance between the two distributions up to a certain threshold $t$ can improve alignment performance. To examine this, we propose the perturbation method to control the distance between the two distributions, represented by $\Delta\mu$.

### 2.2.2 DCPO-p: PERTURBATION METHOD

While using a captioning model is an effective way to address the *conflict distribution*, it risks deviating from the original distribution of prompt $c$, and the distributions of preferred and less preferred images may still remain close. To tackle these issues, we propose a perturbation method. In this approach, we assume that prompt $c$ is highly relevant to the preferred image $x_0^w$ and aim to generate a less relevant caption, denoted as $c_p$, based on prompt $c$. To achieve this, we use the model $W_\phi(c_p|c)$, which generates a perturbed version of prompt $c$, altering its semantic meaning. In this framework, prompt $c$ corresponds to the preferred caption $z^w$ ($c = z^w$), while the perturbed prompt

$c_p$ represents the less-preferred caption $z^l$ ($c_p = z^l$). For the perturbation model $W_\phi$, we utilized the DIPPER model (Krishna et al., 2024) built by fine-tuning the T5-XXL (Chung et al., 2022) to produce a degraded version of the prompt $c$.

We define three levels of perturbation: **1) Weak:** where prompt $c_p$ has high semantic similarity to prompt $c$, with minimal differences. **2) Medium:** where the semantic difference between prompt $c_p$ and $c$ is more pronounced than in the weak level. **3) Strong:** where the majority of the semantics in prompt $c_p$ differ significantly from prompt $c$. For further details on the perturbation method, see Appendix C.

The main advantage of DCPO-p is to reduce the captioning process cost while staying closer to the original data distribution by using prompt $c$ as the preferred caption. However, we observe that the quality of captions in DCPO-c outperforms that of the original prompt $c$, as shown in Table 5 in Appendix B. Based on this observation, we propose a hybrid method to improve the alignment performance by combining captioning and perturbation techniques.

### 2.2.3 DCPO-H: HYBRID METHOD

In this method, instead of perturbing the prompt $c$, we perturb the caption $z$ generated by the model $Q_\phi(z|x, c)$ based on the image $x$ and prompt $c$. As discussed in Section 2.2.1, the goal of the perturbation method is to increase the distance between the two distributions. However, the correlation between the image $x_0$ and prompt $c$ significantly impacts alignment performance. Therefore, we propose Hypothesis 2.

**Hypothesis 2.** *Let $S(c, x)$ represent the correlation score between prompt $c$ and image $x$, and $P(p_\theta(c_1, c_2))$ denote the performance of model $p_\theta$ optimized on captions $c_1$ and $c_2$ with DCPO, where $W_\phi$ is the perturbation model. If $S(z, x) > S(c, x)$, then $P(p_\theta(z^w, z_p^w \sim W_\phi(z_p^w|z^w))) > P(p_\theta(c, c_p \sim W_\phi(c_p|c)))$.*

In Section 3.3, we provide experimental evidence supporting Hypothesis 2 and investigate the potential of using $z_p^l \sim W_\phi(z_p^l|z^l)$ as the less-preferred caption $z^l$, instead of $z_p^w \sim W_\phi(z_p^w|z^w)$ as originally proposed in Hypothesis 2.

## 3 EXPERIMENTS

We fine-tuned the U-Net model of Stable Diffusion 2.1 (SD 2.1) using DCPO on the *Pick-Double Caption* dataset and compared it to SD 2.1 models fine-tuned with SFT$_{\text{Chosen}}$, Diffusion-DPO, and MaPO on Pick-a-Pic v2 across various metrics. We first describe the *Pick-Double Caption* dataset and compare it to Pick-a-Pic v2. Subsequently, we provide an in-depth analysis of the results. For further details on the fine-tuning, refer to Appendix D.

Table 1: Results on PickScore, HSPv2.1, ImageReward (normalized), and CLIPScore. We show that DCPO significantly improves on Pickscore, HSP2.1, and ImageReward.

| | Pick Score (↑) | HPSv2.1 (↑) | Image Reward (↑) | CLIP Score (↑) |
|---|---|---|---|---|
| *Results from other methods* | | | | |
| SD 2.1 | 20.30 | 25.17 | 55.8 | 26.84 |
| SFT$_{\text{Chosen}}$ | 20.35 | 25.09 | 56.4 | 26.98 |
| Diffusion-DPO | 20.36 | 25.10 | 56.4 | 26.98 |
| MaPO | 19.41 | 24.47 | 50.4 | 24.82 |
| *Results from our methods* | | | | |
| DCPO-c (LLaVA) | _20.46_ | 25.10 | 56.5 | _27.00_ |
| DCPO-c (Emu2) | _20.46_ | 25.06 | _56.6_ | 26.97 |
| DCPO-p | 20.28 | _25.42_ | 54.2 | 26.98 |
| DCPO-h (LLaVA) | **20.57** | **25.62** | **58.2** | **27.13** |

### 3.1 PICK-DOUBLE CAPTION DATASET

Motivated by the *conflict distribution* observed in previous preference datasets, we applied the captioning method described in Section 2.2.1 to generate unique captions for each image in the Pick-a-Pic v2 dataset. For the *Pick-Double Caption* dataset, we sampled 20,000 instances from Pick-a-Pic v2 and cleaned the samples as detailed in Appendix B. We then employed two state-of-the-art captioning models, LLaVa-1.6-34B and Emu2-37B, to generate captions for both the preferred and less preferred images, as illustrated in Figure 2.

To generate the captions, we used two different prompting strategies: 1) **Conditional prompt:** where the model was explicitly instructed to generate a caption for image $x$ based on the given prompt $c$, and 2) **Non-conditional prompt:** where the model provided a general description of the image in one sentence without referring to a specific prompt. More details are in Appendix B.

Table 2: Results on the GenEval Benchmark. DCPO significantly enhances model performance in generating the correct number of objects, improving image quality in terms of colors, and constructing attributes accurately.

| Method | Overall | Single object | Two objects | Counting | Colors | Position | Attribute binding |
|---|---|---|---|---|---|---|---|
| *Results from other methods* | | | | | | | |
| SD 2.1[2] | 0.4775 | 0.96 | **0.52** | 0.35 | 0.80 | **0.09** | 0.15 |
| SFT$_{Chosen}$ | 0.4797 | **1.00** | 0.42 | 0.42 | 0.81 | 0.07 | 0.14 |
| Diffusion-DPO | 0.4857 | 0.99 | 0.48 | 0.46 | 0.83 | 0.04 | 0.11 |
| MaPO | 0.4106 | 0.98 | 0.40 | 0.28 | 0.66 | 0.06 | 0.09 |
| *Results from our methods* | | | | | | | |
| DCPO-c (LLaVA) | 0.4971 | **1.00** | 0.43 | 0.53 | **0.85** | 0.02 | 0.14 |
| DCPO-c (Emu2) | 0.4925 | **1.00** | 0.41 | 0.50 | **0.85** | 0.04 | 0.15 |
| DCPO-p | 0.4906 | **1.00** | 0.41 | 0.50 | 0.83 | 0.03 | **0.17** |
| DCPO-h (LLaVA) | **0.5100** | 0.99 | 0.51 | **0.54** | 0.84 | 0.05 | 0.14 |

We evaluated the captions generated by LLaVA and Emu2 using CLIPscore, which revealed several key insights. LLaVA produced captions that have more correlation with the images for both preferred and less preferred samples compared to Emu2 and the original captions, although LLaVA's captions were significantly longer (see Table 5 in Appendix B). Models fine-tuned on captions from the conditional prompt strategy outperformed those using the non-conditional approach, though the conditional prompt captions were twice as long. Interestingly, despite Emu2 generating much shorter captions, the models fine-tuned on Emu2 were comparable to those fine-tuned on the original prompts from Pick-a-Pic v2.

A key challenge is generating captions for the less preferred images using the captioning method. We observed that in both prompting strategies, the captions for the preferred images are more aligned with the original prompt $c$ distribution. However, the non-conditional prompt strategy often produces captions for less preferred images that are out-of-distribution (OOD) from the original prompt $c$ in most cases. We will explore this further in Section 3.3.

Finally, we observe that the key advantage of the *Pick-Double Caption* dataset is the greater difference in CLIPscore ($\Delta\mu$) between preferred and less preferred images compared to the original prompts. Specifically, while the original prompt has a $\Delta\mu$ of **1.3**, LLaVA shows a much larger difference at **4.3**, and Emu2 at **2.8**. This increased gap reflects improved alignment performance in models fine-tuned on this dataset, indicating that the captioning method mitigates the *conflict distribution*.

## 3.2 PERFORMANCE COMPARISONS

We evaluated all methods on 2,500 unique prompts from the Pick-a-Pic v2 (Kirstain et al., 2023) dataset, measuring performance using Pickscore (Kirstain et al., 2023), CLIPscore (Hessel et al., 2022), and Normalized ImageReward (Xu et al., 2023). Additionally, we generated images from 3,200 prompts in the HPSv2 (Wu et al., 2023b) benchmark and evaluated them using the HPSv2.1 model. To provide a comprehensive evaluation, we also compared the methods using GenEval (Ghosh et al., 2023), focusing on how well the fine-tuned models generated images with the correct number of objects, accurate colors, and proper object positioning.

We compared different versions of DCPO, including the captioning (DCPO-c), perturbation (DCPO-p), and hybrid (DCPO-h) methods, with other approaches, as outlined in Section 2.2. For more information on the fine-tuning process of the models, refer to Appendix D.

The results in Tables 1 and 2 show that DCPO-h significantly outperforms the best scores from other methods, with improvements of **+0.21** in Pickscore, **+0.45** in HPSv2.1, **+1.8** in ImageReward, **+0.15** in CLIPscore, and **+3%** in GenEval. Additionally, the results demonstrate that DCPO-c outperforms all other methods on GenEval, Pickscore, and CLIPscore. While DCPO-p performs slightly worse than DCPO-c, it still exceeds SD 2.1, SFT, Diffusion-DPO, and MaPO on GenEval. However, its scores on ImageReward and Pickscore suggest that it underperforms compared to the

---

[2]Note that we rerun all the models on same seeds to have a fair comparison.

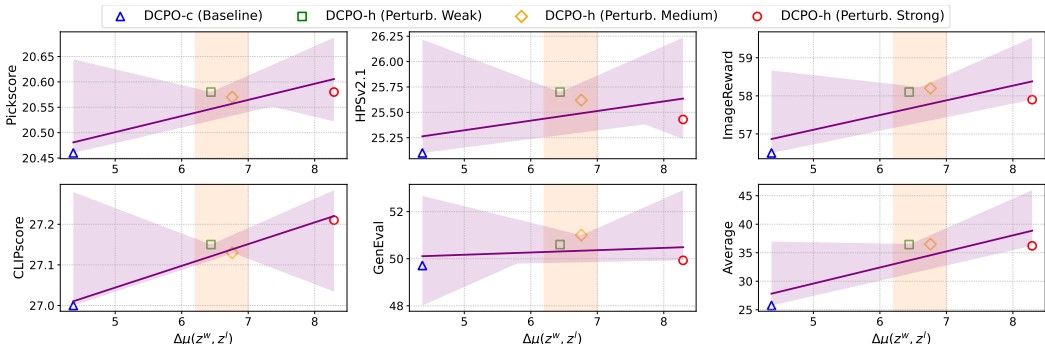

Figure 5: Performance comparison of DCPO-c and DCPO-h on different perturbation levels. We plotted regression lines for the four models, showing that as $\Delta\mu$ increases, performance improves but drops after a threshold $t$ (orange boundary).

Table 3: Performance comparison of DCPO-h and DCPO-p across different perturbation levels. The perturbation method has a strong impact on captions that are more closely correlated with images.

| Method | Pair Caption | Perturbed Level | Pickscore (↑) | HPSv2.1 (↑) | ImageReward (↑) | CLIPscore (↑) | GenEval (↑) |
|--------|-------------|-----------------|---------------|-------------|-----------------|---------------|-------------|
| DCPO-p | $(c, c_p)$ | weak | 20.28 | 25.42 | 54.20 | 26.98 | 0.4906 |
| DCPO-h | $(z^w, z_p^w)$ | weak | 20.55 | 25.61 | 57.70 | 27.07 | **0.5070** |
| DCPO-h | $(z^w, z_p^l)$ | weak | **20.58** | **25.70** | **58.10** | **27.15** | 0.5060 |
| DCPO-p | $(c, c_p)$ | medium | 20.21 | 25.34 | 53.10 | 26.87 | 0.4852 |
| DCPO-h | $(z^w, z_p^w)$ | medium | **20.59** | **25.73** | **58.47** | 27.12 | 0.5008 |
| DCPO-h | $(z^w, z_p^l)$ | medium | 20.57 | 25.62 | 58.20 | **27.13** | **0.5100** |
| DCPO-p | $(c, c_p)$ | strong | 20.31 | 25.06 | 54.60 | 27.03 | 0.4868 |
| DCPO-h | $(z^w, z_p^w)$ | strong | 20.57 | 25.27 | 57.43 | 27.18 | **0.5110** |
| DCPO-h | $(z^w, z_p^l)$ | strong | **20.58** | **25.43** | **57.90** | **27.21** | 0.4993 |

other approaches. Importantly, DCPO-p shows significant improvement over the other methods on HPSv2.1, highlighting the effectiveness of the perturbation method.

## 3.3 Ablation Studies and Analysis

**Support of Hypothesis 1.** As described in Section 2.2.2, we defined three levels of perturbation: weak, medium, and strong. In Hypothesis 1, we proposed that increasing the distance between the distributions of preferred and less preferred images $\Delta\mu$ improves model alignment performance. To explore this, we fine-tuned SD 2.1 using the DCPO-h method with three levels of perturbation applied to the less preferred captions $z^l$ generated by LLaVA. The results in Figure 5 show that increasing the distance $\Delta\mu$ between the two distributions enhances performance. However, this distance must be controlled and kept below a threshold $t$, a hyperparameter that may vary depending on the task. These findings support our hypothesis.

**Support of Hypothesis 2.** To illustrate the impact of the correlation between the prompt $c$ and image $x$ on the perturbation method, we perturbed both the original prompt $c$ and the less preferred caption $z^w$, generated by the model $Q_\phi$, where $z^w \sim W_\phi(z^w|Q_\phi(z^w|x^w, c))$. At the same time, we kept the caption generated by $Q_\phi$ for the preferred image as the preferred caption, $z^w \sim Q(z^w|x^w, c)$. In this case, we assume $Q_\phi = $ LLaVA and $W_\phi = $ DIPPER. The results in Table 5 in Appendix B show that the caption $z$ generated by LLaVA is more correlated with the image $x$ than the original prompt $c$, indicating that $S(z, x) > S(c, x)$. Based on the results in Table 3, we conclude that perturbing more correlated captions leads to better performance.

**In- vs. Out-of Distribution.** We evaluated DCPO on in-distribution and out-of-distribution (OOD) data. As discussed in Section 3.1, the captioning model can generate OOD captions. To explore this, we fine-tuned SD 2.1 with DCPO-h using LLaVA and Emu2 captions at a medium perturbation level. Figure 6 shows that in-distribution data significantly improve alignment perfor-

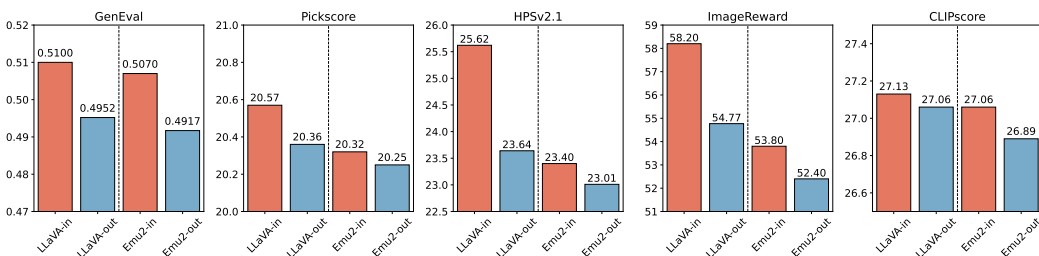

Figure 6: Comparison of DCPO-h performance on in-distribution and out-of-distribution data.

Table 4: Performance comparison of DCPO and Diffusion-DPO fine-tuned on the *Pick-Double Caption* dataset. While larger captions improve the performance of Diffusion-DPO, DCPO-h still significantly outperforms Diffusion-DPO.

| Method | Input Prompt | Token Length (Avg) | Pickscore (↑) | HPSv2.1 (↑) | ImageReward (↑) | CLIPscore (↑) | GenEval (↑) |
|---|---|---|---|---|---|---|---|
| Diffusion-DPO | prompt $c$ | 15.95 | 20.36 | 25.10 | 56.4 | 26.98 | 0.4857 |
| Diffusion-DPO | caption $z^w$ (LLaVA) | 32.32 | 20.40 | 25.19 | 56.6 | 27.10 | 0.4958 |
| Diffusion-DPO | caption $z^w$ (Emu2) | 7.75 | 20.36 | 25.08 | 56.3 | 26.98 | 0.4960 |
| DCPO-h (LLaVA) | Pair ($z^w, z_p^l$) | (32.32, 31.17) | **20.57** | **25.62** | **58.2** | 27.13 | 0.5100 |
| DCPO-h (LLaVA) | Pair ($z^w, z_p^w$) | (32.32, 27.01) | **20.57** | 25.27 | 57.4 | **27.18** | **0.5110** |

mance, while OOD results for LLaVA in GenEval, Pickscore, and CLIPscore are comparable to Diffusion-DPO. Similar behavior was observed for DCPO-c, as noted in Appendix D.

**Effectiveness of the DCPO.** Our analysis shows that LLaVA captions are twice the length of the original prompt $c$, raising the question of *whether DCPO's improvement is due to data quality or the optimization method*. To explore this, we fine-tuned SD 2.1 with Diffusion-DPO using LLaVA and Emu2 captions instead of the original prompt. The results in Table 4 show that models fine-tuned on LLaVA captions outperform Diffusion-DPO with the original prompt. However, DCPO-h still surpasses the new Diffusion-DPO models, demonstrating the effectiveness of the proposed optimization algorithm.

**Explore on $\beta$.** In DCPO, $\beta$ is a key hyperparameter. To evaluate its impact, we fine-tuned SD 2.1 using different values of $\beta = \{500, 1000, 1500, 2500, 5000\}$. Interestingly, in Figure 7 we observed that $\beta = 500$ showed significant improvements on HPSv2.1 and GenEval, even surpassing DCPO-h with $\beta = 5000$, our best-reported model. Additional results for different $\beta$ values can be found in Appendix D.

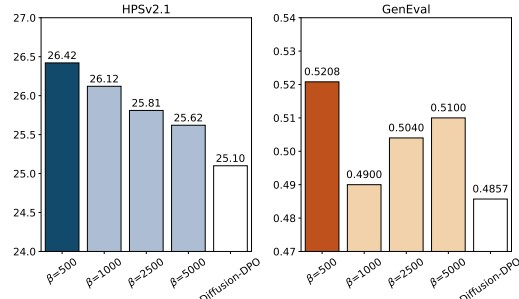

Figure 7: DCPO-h performance comparison across various $\beta$ values, evaluated on HPSv2.1 and GenEval.

**DCPO-h vs Diffusion-DPO on GPT-4o Judgment.** We evaluated DCPO-h and Diffusion-DPO using GPT-4o on the PartiPrompts benchmark, consisting of 1,632 prompts. GPT-4o assessed images based on three criteria: **Q1)** General Preference (*Which image do you prefer given the prompt?*), **Q2)** Visual Appeal (*Which image is more visually appealing?*), and **Q3)** Prompt Alignment (*Which image better fits the text description?*). As shown in Figure 8, DCPO-h outperformed Diffusion-DPO in Q1 and Q2, with win rates of 58% and 66%. To see the style of the prompts, refer to Appendix E.

## 4 RELATED WORKS

**Aligning Diffusion Models.** Recent advances in preference alignment methods of text-to-image diffusion models have shown that reinforcement learning-free (RL-free) methods (Wallace et al.,

Figure 8: **(Left)** PartiPrompts benchmark results for three evaluation questions, as voted by GPT-4o. **(Right)** Qualitative comparison between DCPO-h and Diffusion-DPO fine-tuned on SD 2.1. DCPO-h shows better prompt adherence and realism, with outputs that align more closely with human preferences, emphasizing high contrast, vivid colors, fine detail, and well-focused composition.

2024; Yang et al., 2024; Li et al., 2024; Yuan et al., 2024; Gambashidze et al., 2024; Park et al., 2024) outperforms RL-based approaches (Fan & Lee, 2023; Fan et al., 2023; Hao et al., 2023; Lee et al., 2023; Xu et al., 2023; Prabhudesai et al., 2024; Black et al., 2024; Clark et al., 2024) mainly because they eliminates the need for an explicit reward model. Initially proposed for Large Language Models (LLMs), Direct Preference Optimization (DPO) (Rafailov et al., 2024) reformulate the RLHF objective in a closed-form manner and introduce it as an implicit reward model with a simple classification objective. Diffusion-DPO (Wallace et al., 2024) directly adopts DPO method into text-to-image diffusion models, utilizing pairwise preference datasets consisting of text and images to guide alignment. Diffusion-KTO (Li et al., 2024) incorporates Kahneman & Tversky model of human utility to align these models, simplifying the process by using images with binary feedback signals, i.e., likes or dislikes instead of pairwise preference data. To enhance flexibility, Hong et al. (2024) introduce MaPO, an alignment technique independent of a reference model previously used by other methods, enabling greater control over stylistic adaptations. However, previous alignment methods optimize diffusion models based on a single prompt for a pair of images, which supports the *irrelevant prompts* issue explored in Section 2.1.

**Text-to-image Preference Datasets.** Text-to-image image preference datasets commonly involve the text prompt to generate the images, and two or more images are ranked according to human preference. HPS (Wu et al., 2023c) and HPSv2 (Wu et al., 2023b) create multiple images using a series of image generation models for a single prompt, and the images are ranked according to real-world human preferences. Moreover, a classifier is trained using the gathered preference dataset, which can be used as a metric for image-aligning tasks. Also, Pick-a-Pic v2 (Kirstain et al., 2023) follows a similar structure to create a pairwise preference dataset along with their CLIP (Radford et al., 2021) based scoring function, Pickscore. While these datasets are carefully created, having only one prompt for both or all the images introduces *conflict distribution*, which will be further discussed in Section 2.1. For this reason, we modified the Pick-a-Pic v2 dataset using recaptioning and perturbation methods to improve image alignment performance.

## 5 CONCLUSION AND LIMITATIONS

In this paper, we present a novel preference optimization method for aligning text-to-image diffusion models called Dual Caption Preference Optimization (DCPO). We tackle two major challenges in previous preference datasets and optimization algorithms: the *conflict distribution* and *irrelevant prompt*. To overcome these issues, we introduce the *Pick-Double Caption* dataset, a modified version of the Pick-a-Pic v2 dataset. We also identify difficulties in generating captions, particularly the risk of out-of-distribution captions for images, and propose three approaches: 1) captioning (DCPO-c), 2) perturbation (DCPO-p), and 3) a hybrid method (DCPO-h). Our results show that DCPO-h significantly enhances alignment performance, outperforming methods like MaPO and Diffusion-DPO across multiple metrics.

**Limitations.** Although DCPO shows strong performance across various metrics, the captioning and perturbation methods are resource-intensive. We encourage future research to explore cost-effective alternatives to these methods. Additionally, the potential of using different backbones, such as Stable Diffusion XL (SDXL) (Rombach et al., 2022), has not been explored in the context of DCPO. We also invite researchers to investigate DCPO's effectiveness on other tasks, such as safety. We believe our work will have a significant impact on the alignment research community.

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

# A BACKGROUND

## A.1 DIFFUSION MODELS

Based on samples from a data distribution $q(x_0)$, a noise scheduling function $\alpha_t$ and $\sigma_t$ (Rombach et al., 2022) denoising diffusion models (Song et al., 2020b) are generative models $p_\theta(x_0)$ that operate through a discrete-time reverse process structured as a Markov Decision Proces where

$$p_\theta(x_{t-1}|x_t) = \mathcal{N}(x_{t-1}; \mu_\theta(x_t), \sigma_{t|t-1}^2 \frac{\sigma_{t-1}^2}{\sigma_t^2} I). \tag{3}$$

The training process involves minimizing the evidence lower bound (ELBO) associated with this model (Song et al., 2021):

$$L_{DM} = \mathbb{E}_{x_0,\epsilon,t,x_t}[\omega(\lambda_t)||\epsilon - \epsilon_\theta(x_t,t)||_2^2] \tag{4}$$

where $\epsilon \sim \mathcal{N}(0, I)$, $t \sim \mathcal{U}(o,T)$, $x_t$ $q(x_t|x_0) = \mathcal{N}(x_t; \alpha_t x_0, \sigma_t^2 I).\lambda_t = \alpha_t^2/\sigma_t^2$ is a signal-to-noise ratio (Kingma et al., 2021), $\omega(\lambda_t)$ is a predefined weighting function (Song & Ermon, 2019).

## A.2 PREFERENCE OPTIMIZATION

Aligning a generative model typically involves fine-tuning it to produce outputs that are more aligned with human preferences. Estimating the reward model $r$ based on human preference is generally challenging, as we do not have direct access to the reward model. However, if we assume the availability of ranked data generated under a given condition $c$, where $x_0^w \succ x_0^l|c$ (with $x_0^w$ representing the preferred sample and $x_0^l$ the less-preferred sample), we can apply the Bradley-Terry theory to model these preferences. The Bradley-Terry (BT) model expresses human preferences as follows:

$$p_{BT}(x_0^w \succ x_0^l|c) = \sigma(r(c, x_0^w) - r(c, x_0^l)) \tag{5}$$

where $\sigma$ denotes the sigmoid function, and $r(x_0, c)$ is derived from a neural network parameterized by $\phi$, which is estimated through maximum likelihood training for binary classification as follows:

$$L_{BT}(\phi) = -\mathbb{E}_{c,x_0^w,x_0^l}[\log \sigma(r_\phi(c,x_0^w) - r_\phi(c,x_0^l))] \tag{6}$$

where the prompt $c$ and data pairs $x_0^w$, $x_0^l$ are sourced from a dataset that humans have annotated.

This approach to reward modeling has gained popularity in aligning large language models, particularly when combined with reinforcement learning (RL) techniques like proximal policy optimization (PPO) (Schulman et al., 2017) to fine-tune the model based on rewards learned from human preferences, known as Reinforcement Learning from Human Feedback (RLHF) (Ouyang et al., 2022). The goal of RLHF is to optimize the conditional distribution $p(x_0|c)$ (where $c \sim D_c$) such that the reward model $r(c, x_0)$ is maximized, while keeping the policy model within the desired distribution using a KL-divergence term to ensure it remains reachable under the following objective:

$$\max_{p_\theta} \mathbb{E}_{c\sim\mathcal{D}_c, x_0 \sim p_\theta(x_0|c)}[r(c, x_0)] - \beta\mathbb{D}_{KL}[p_\theta(x_0|c)||p_{\text{ref}}(x_0|c)] \tag{7}$$

where $\beta$ controls how far the policy model $p_\theta$ can deviate from the reference model $p_{ref}$. It can be demonstrated that the objective in Equation 7 converges to the following policy model:

$$p_\theta^*(x_0|c) = p_{\text{ref}}(x_0|c)\exp(r(c, x_0)/\beta)/Z(c) \tag{8}$$

where $Z$ is the partition function, the training objective for $p_\theta$, inspired by DPO, has been derived to be equivalent to Equation 8 without the need for an explicit reward model $r(x, c)$. Instead, it learns directly from the preference data $(c, x_0^w, x_0^l) \sim D$.

$$L_{\text{DPO}}(\theta) = -\mathbb{E}_{c,x_0^w,x_0^l}[\log \sigma(\beta \log \frac{p_\theta(x_0^w|c)}{p_{\text{ref}}(x_0^w|c)} - \beta \log \frac{p_\theta(x_0^l|c)}{p_{\text{ref}}(x_0^l|c)})] \qquad (9)$$

where $\sigma$ represents the sigmoid function.

Through this reparameterization, instead of first optimizing the reward function $r$ and then applying reinforcement learning, the method directly optimizes the conditional distribution $p_\theta(x_0|c)$.

## B  PICK-DOUBLE CAPTION DATASET

In this section, we provide details about the *Pick-Double Caption* dataset. As discussed in Section 3.1, we sampled 20,000 instances from the Pick-a-Pic v2 dataset and excluded those with equal preference scores. We plot the distribution of the original prompts, as shown in Figure 9.

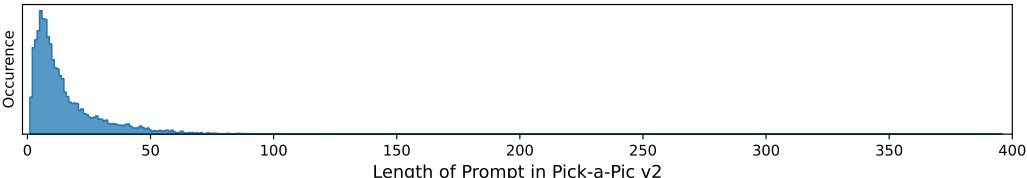

Figure 9: Token distribution of original prompt.

We observed that some prompts contained only one or two words, while others were excessively long. To ensure a fair comparison, we removed prompts that were too short or too long, leaving us with approximately 17,000 instances. We then generated captions using two state-of-the-art models, LLaVA-1.6-34B, and Emu2-32B. Figure 10 provides examples from the dataset.

As explained in Section 3.1, we utilized two types of prompts to generate captions: 1) Conditional prompt and 2) Non-conditional prompt. Below, we outline the specific prompts used for each captioning method.

> **Example of Conditional Prompt**
>
> Using one sentence, describe the image based on the following prompt: *playing chess tournament on the moon.*

> **Example of Non-Conditional Prompt**
>
> Using one sentence, describe the image.

Table 5 presents a statistical analysis of the *Pick-Double Caption* dataset. With the non-conditional prompt method, we found that the average token length of captions generated by LLaVA is similar to that of the original prompts. However, captions generated by LLaVA using conditional prompts are twice as long as the original prompts. Additionally, Emu2 generated captions that, on average, are half the length of the original prompts for both methods.

## C  MORE DETAILS ON PERTURBATION METHOD

We provide the setups for the LLM-based perturbation process involved in the DCPO-p and DCPO-h pipelines. Similarly to the method of constructing paraphrasing adversarial attacks as synonym-swapping perturbation by Krishna et al. (2024), we use DIPPER (Krishna et al., 2024), a text generation model built by fine-tuning T5-XXL (Chung et al., 2022), to create semantically perturbed captions or prompts, as shown in Table 6. Our three levels of perturbation are achieved by only altering the setting of lexicon diversity (0 to 100) in DIPPER - we use 40 for **Weak**, 60 for **Medium**,

Table 5: Statistical information on the Pick-Double Caption dataset, including the CLIPscore of in-distribution data and average token count of captions generated by LLaVA and Emu2 for both in-distribution and out-of-distribution data.

| Text | Token Len. (Avg-in) | Token Len. (Avg-out) | CLIP score (in) | CLIP score (out) |
|---|---|---|---|---|
| prompt $c$ | 15.95 | 15.95 | (26.74, 25.41) | (26.74, 25.41) |
| caption $z^w$ (LLaVA) | 32.32 | 17.69 | 30.85 | 29.04 |
| caption $z^l$ (LLaVA) | 32.83 | 17.91 | 26.48 | 28.29 |
| caption $z^w$ (Emu2) | 7.75 | 8.40 | 25.44 | 25.18 |
| caption $z^l$ (Emu2) | 7.84 | 8.44 | 22.64 | 24.88 |

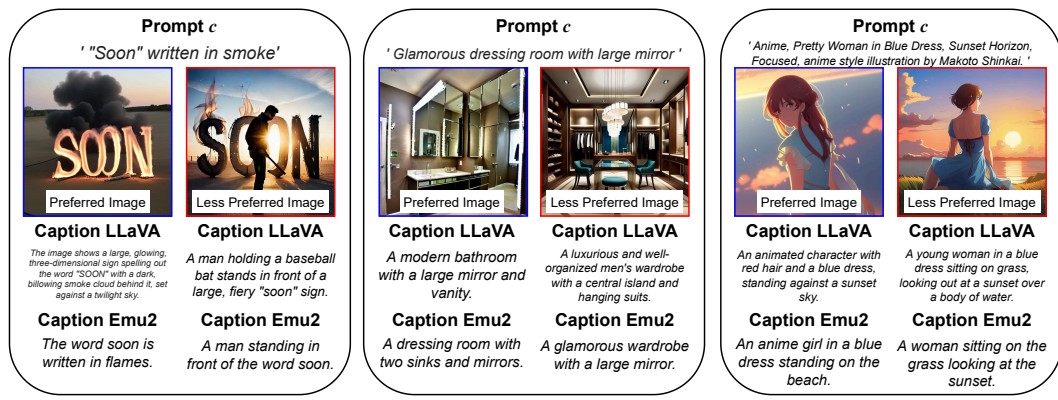

Figure 10: Examples of Pick-Double Caption dataset.

and 80 for **Strong**. We also use *"Text perturbation for variable text-to-image prompt."* to prompt the perturbation. We hereby provide a code snippet to showcase the whole process to perturb a sample input:

```python
from transformers import T5Tokenizer, T5ForConditionalGeneration
class DipperParaphraser(object):
    # As defined in https://huggingface.co/kalpeshk2011/dipper-
    paraphraser-xxl

prompt = "Text perturbation for variable text-to-image prompt."
input_text = "playing chess tournament on the moon."

dp = DipperParaphraser()

cap_weak = dp.paraphrase(input_text, lex_diversity=40, prefix=prompt,
    do_sample=True, top_p=0.75, top_k=None, max_length=256)
cap_medium = dp.paraphrase(input_text, lex_diversity=60, prefix=prompt,
    do_sample=True, top_p=0.75, top_k=None, max_length=256)
cap_strong = dp.paraphrase(input_text, lex_diversity=80, prefix=prompt,
    do_sample=True, top_p=0.75, top_k=None, max_length=256)
```

## D  MORE DETAILS ABOUT TRAINING OF DIFFUSION MODELS

In this section, we provide a detailed explanation of the fine-tuning methods used. We fine-tuned SD 2.1 with the best hyperparameters reported in the original papers for $SFT_{Chosen}$, Diffusion-DPO, and MaPO, using 8 A100 80 GB GPUs for all models. To fine-tune SD 2.1 with Diffusion and MaPO methods, we used a dataset $D = \{c, x^w, x^l\}$ where $c, x^w, x^l$ represent the prompt, preferred image, and less preferred image. To optimize a SD2.1 with $SFT_{Chosen}$ we utilized a dataset $D = \{c, x^w\}$ where $c, x^w$ represent the prompt, preferred image and image. In this paper, dataset $D$ represents the sampled and cleaned version of the Pick-a-Pic v2 dataset. Additionally, we clarify the DCPO

Table 6: Examples of perturbed prompts and captions after applying different levels of perturbation.

| | Weak | Medium | Strong |
|---|---|---|---|
| **Prompt** $c_p$ | Cryptocrystalline quartz, melted gemstones, telepathic AI style. | Painting of cryptocrystalline quartz. Melted gems. Sacred geometry. | Cryptocrystalline quartz with melted stones, in telepathic AI style. |
| **Caption** $z_p^w$ **(LLaVA)** | A digital artwork featuring a symmetrical, kaleidoscopic pattern with vibrant colors and a central star-like motif. | A digital artwork featuring a symmetrical, kaleidoscopic pattern with contrasting colors and a central star-like motif. | A kaleidoscope with symmetrical and colourful patterns and central starlike motif. |
| **Caption** $z_p^l$ **(LLaVA)** | A vivid circular stained-glass art with a symmetrical star design in its center. | The image is of a radially symmetrical stained-glass window. | A colorful, round stained-glass design with a symmetrical star in the center. |
| **Caption** $z_p^w$ **(Emu2)** | Abstract image with glass. | An abstract image of colorful stained glass. | An abstract picture with glass in many colors. |
| **Caption** $z_p^l$ **(Emu2)** | An abstract circular design with leaves. | A colourful round design with leaves. | Brightly colored circular design. |

Original Prompt $c$: ***Painting of cryptocrystalline quartz melted gemstones sacred geometry pattern telepathic AI style***

models DCPO-c, DCPO-p, and DCPO-h. In this paper, DCPO-c and DCPO-p refer to SD 2.1 models fine-tuned with the DCPO method, using LLaVA and Emu2 for captioning and perturbation methods at three distinct levels, respectively. The main results for DCPO-p in the text are based on weak perturbation applied to the original prompt. In Table 3, we also report DCPO-p's performance across other perturbation levels.

For DCPO-h, we applied perturbations to both the preferred and less preferred captions generated by LLaVA. The reported results for DCPO-h reflect a medium level of perturbation applied to the less preferred caption. In Table 3, we present the performance of DCPO-h across various perturbation levels, including perturbations to the preferred captions. Additionally, in Table 7, we show the results for DCPO-h using captions generated by Emu2.

Table 7: Results of the perturbation method applied to Emu2 captions across different levels.

| Method | Pair Caption | Perturbed Level | Pickscore ($\uparrow$) | HPSv2.1 ($\uparrow$) | ImageReward ($\uparrow$) | CLIPscore ($\uparrow$) | GenEval ($\uparrow$) |
|---|---|---|---|---|---|---|---|
| DCPO-h | $(z^w, z_p^w)$ | weak | 20.10 | 21.23 | 49.7 | 26.87 | 0.5003 |
| DCPO-h | $(z^w, z_p^l)$ | weak | 20.32 | 23.4 | 53.8 | 27.06 | 0.5070 |
| DCPO-h | $(z^w, z_p^w)$ | medium | 20.31 | 23.08 | 53.2 | 27.01 | 0.4895 |
| DCPO-h | $(z^w, z_p^l)$ | medium | 20.33 | 23.22 | 53.8 | 27.09 | 0.5009 |
| DCPO-h | $(z^w, z_p^w)$ | strong | 20.31 | 22.95 | 53.1 | 27.11 | 0.4878 |
| DCPO-h | $(z^w, z_p^l)$ | strong | 20.35 | 23.24 | 53.63 | 27.08 | 0.5050 |

The key findings indicate that perturbation on short captions not only fails to improve performance but also produces worse outcomes compared to DCPO-c (Emu2).

Additionally, we conducted more experiments on in-distribution and out-of-distribution data. For this, we generated out-of-distribution data using LLaVA and Emu2 in the captioning setup. As shown in Figure 11, in-distribution data generally outperformed out-of-distribution data. However, the most significant improvement was observed with the hybrid method, as reported in Figure 6.

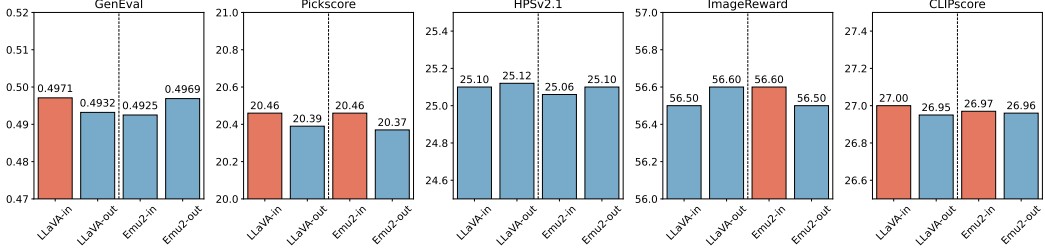

Figure 11: Comparison of DCPO-c performance on in-distribution and out-of-distribution data.

Table 8 presents the performance details for different values of $\beta$, conducted using the medium level of DCPO-h. The results indicate that while lower values of $\beta$ significantly improve GenEval and

HPSv2.1 on average, the optimal value for $\beta$ is 5000. We suggest that this hyperparameter may vary based on the dataset and task.

Table 8: Results of DCPO-h across different $\beta$.

| Method | $\beta$ | Pickscore ($\uparrow$) | HPSv2.1 ($\uparrow$) | ImageReward ($\uparrow$) | CLIPscore ($\uparrow$) | GenEval ($\uparrow$) |
|--------|---------|------------------------|----------------------|--------------------------|------------------------|----------------------|
| DCPO-h | 500 | 20.43 | **26.42** | 58.1 | 27.02 | **0.5208** |
| DCPO-h | 1000 | 20.51 | 26.12 | **58.2** | 27.10 | 0.4900 |
| DCPO-h | 2500 | 20.53 | 25.81 | 58.0 | 27.02 | 0.5036 |
| DCPO-h | 5000 | **20.57** | 25.62 | **58.2** | **27.13** | 0.5100 |

# E    GPT-4O AS AN EVALUATOR

To obtain binary preferences from the API evaluator, we followed the approach outlined in the MaPO paper (Hong et al., 2024). Similar to Diffusion-DPO, we used three distinct questions to evaluate the images generated by the DCPO-h and Diffusion-DPO models, both utilizing SD 2.1 as the backbone. These questions were presented to the GPT-4o model to identify the preferred image. Below, we provide details of the prompts used.

---

**GPT-4o Evaluation Prompt for Q1: General Preference**

Select the output (a) or (b) that best matches the given prompt. Choose your preferred output, which can be subjective. Your answer should ONLY contain: Output (a) or Output (b).

## Prompt:
`{prompt}`

## Output (a):
The first image attached.

## Output (b):
The second image attached.

## Which image do you prefer given the prompt?

---

**GPT-4o Evaluation Prompt for Q2: Visual Appeal**

Select the output (a) or (b) that best matches the given prompt. Choose your preferred output, which can be subjective. Your answer should ONLY contain: Output (a) or Output (b).

## Prompt:
`{prompt}`

## Output (a):
The first image attached.

## Output (b):
The second image attached.

## Which image is more visually appealing?

---

---

**GPT-4o Evaluation Prompt for Q3: Prompt Alignment**

Select the output (a) or (b) that best matches the given prompt. Choose your preferred output, which can be subjective. Your answer should ONLY contain: Output (a) or Output (b).

## Prompt:
{prompt}

## Output (a):
The first image attached.

## Output (b):
The second image attached.

## Which image better fits the text description?

---

# F ADDITIONAL GENERATION SAMPLES

We also present additional samples for qualitative comparison generated by SD 2.1, SFT$_{Chosen}$, Diffusion-DPO, MaPO, and DCPO-h from prompts on Pickscore, HPSv2, and GenEval benchmarks.

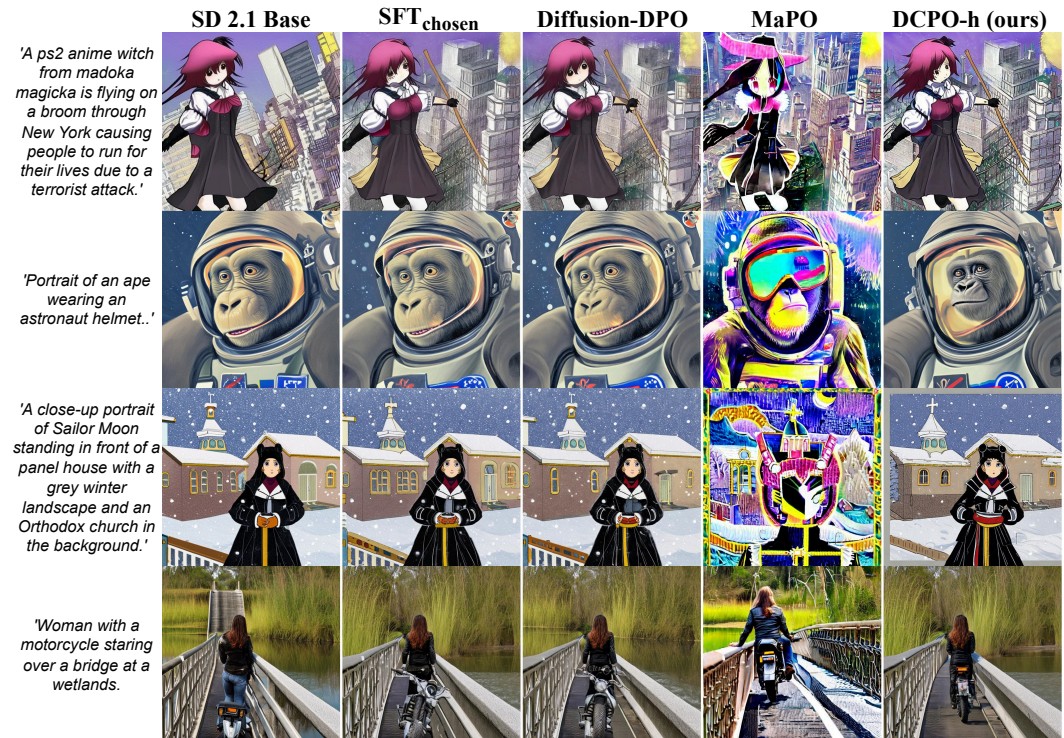

Figure 12: Additional generated outcomes using prompts from HPSv2 benchmark.

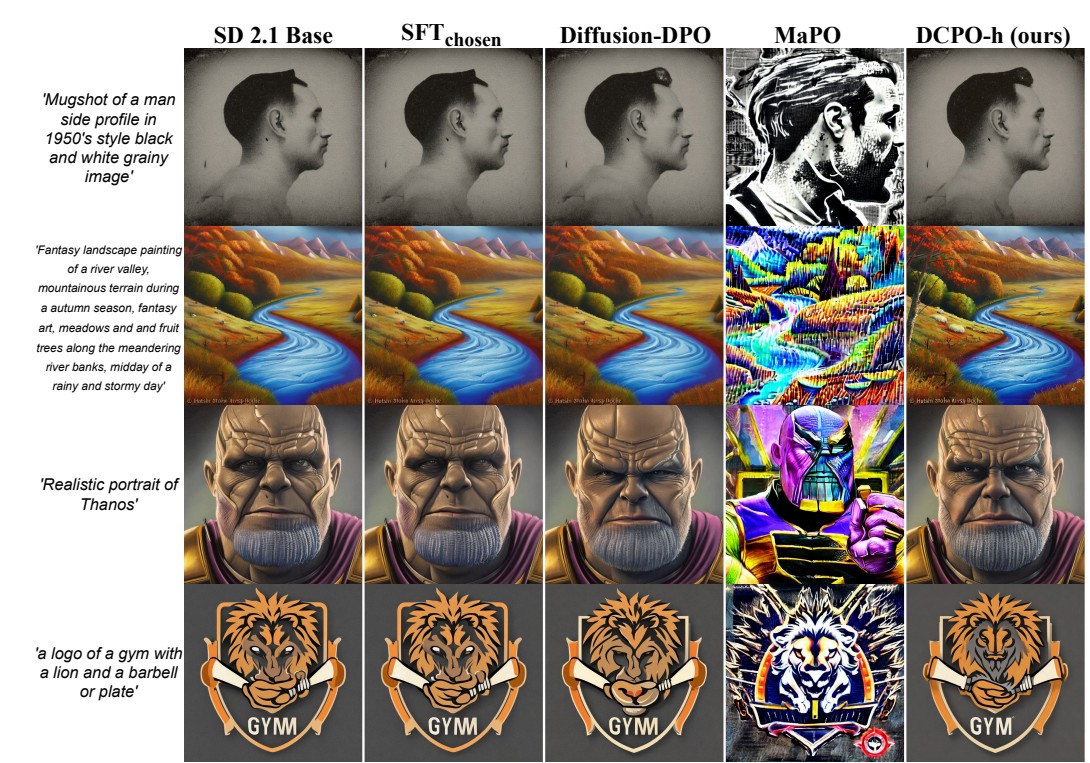

Figure 13: Additional generated outcomes using prompts from Pickscore benchmark.

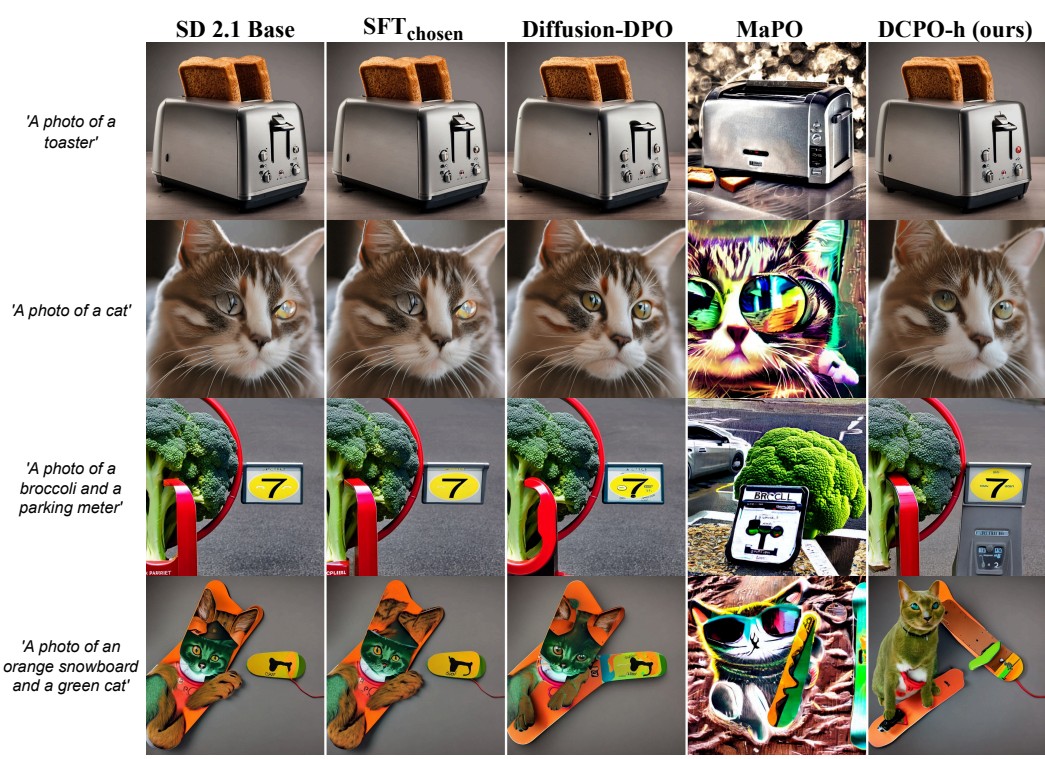

Figure 14: Additional generated outcomes using prompts from GenEval benchmark.

