# OpenReview forum: "Dual Caption Preference Optimization for Diffusion Models"
_ICLR.cc/2025/Conference — ICLR 2025 Conference Withdrawn Submission_

### Official Review · Reviewer_Y9en · 2024-10-30

**Soundness:** 3
**Presentation:** 3
**Contribution:** 2
**Rating:** 6
**Confidence:** 2

**Summary:**

This paper introduces Dual Caption Preference Optimization (DCPO) to enhance text-to-image diffusion models by aligning them with human preferences. Traditional methods face issues like overlapping distributions and irrelevant prompts. DCPO addresses these using two distinct captions for each image, mitigating conflicts in preference data. The authors introduce the Pick-Double Caption dataset to support this approach. They apply three strategies—captioning, perturbation, and hybrid methods—to generate unique captions. Experiments show DCPO improves image quality and prompt relevance. DCPO outperforms prior models on multiple metrics, validating its effectiveness.

**Strengths:**

As a reviewer from a broader field, I am not very familiar with the specific domain of this paper. Therefore, I am reviewing this paper from a generalist’s perspective. The strengths of this paper are:

1. It provides sufficient theoretical support for the motivation, which aligns well with the characteristics of ICLR papers.
2. The issues raised seem quite reasonable.
3. Extensive quantitative and qualitative experiments support the arguments presented.

**Weaknesses:**

However, I still have a few concerns:

1. The issues of conflict distribution and irrelevant prompts seem like two aspects of the same problem—both involve a single prompt (C) corresponding to two different images, which can lead to unstable optimization. Therefore, I think they could be consolidated into a single issue.
2. When comparing generated images, the improvements achieved by the proposed method could be highlighted more clearly; otherwise, it’s often not immediately obvious, as in Figure 1.
3. In fact, the explanations of conflict distribution and irrelevant prompts in the abstract and introduction are quite obscure and difficult to understand. I had to reread these sections several times, only gaining clarity after reading the methods section. This part may need reorganization.

**Questions:**

The issues of conflict distribution and irrelevant prompts seem like two aspects of the same problem—both involve a single prompt (C) corresponding to two different images, which can lead to unstable optimization. Therefore, I think they could be consolidated into a single issue.

---

> ### Author Response · Authors · 2024-11-19
> **Official Reply to Reviewer Y9en**
>
> We greatly value the reviewer’s insightful feedback on our paper and look forward to engaging in a productive discussion, as there is still significant time left in the discussion period. Below, we offer a detailed response to your comments:
>
> ---
> > **W1:** The issues of conflict distribution and irrelevant prompts seem like two aspects of the same problem—both involve a single prompt (C) corresponding to two different images, which can lead to unstable optimization. Therefore, I think they could be consolidated into a single issue.
> ---
>
> **Reply to W1:** We appreciate the reviewer’s observation, but conflict distribution and irrelevant prompts address distinct challenges.
>
> Conflict Distribution refers to the similarity between preferred ($ x^w $) and less preferred ($ x^l $) images, quantified using the correlation between images ($ x $) and captions ($ z $) via CLIPscore. High similarity can make it harder to distinguish preferences during optimization.
>
> Irrelevant Prompts highlight misalignment between the prompt ($ c $) and the less preferred image ($ x^l $). Prompts often include details relevant to $ x^w $ but not $ x^l $, which hinders the model's ability to learn effectively.
>
> For example, even when $ x^w $ and $ x^l $ have similar CLIPscores, a prompt $ c $ may still lack relevance for $ x^l $, introducing additional optimization challenges. These two issues complement but do not fully overlap, as both similarity and prompt relevance impact performance differently.
>
> ---
> > **W2:** When comparing generated images, the improvements achieved by the proposed method could be highlighted more clearly; otherwise, it’s often not immediately obvious, as in Figure 1.
> ---
>
> **Reply to W2:** Thank you for this comment. We have updated Figure 1 in the revised version. Additionally, the reviewer can find more qualitative examples across different benchmarks in Appendix F.
>
> ---
> > **W3:** In fact, the explanations of conflict distribution and irrelevant prompts in the abstract and introduction are quite obscure and difficult to understand. I had to reread these sections several times, only gaining clarity after reading the methods section. This part may need reorganization.
> ---
>
> **Reply to W3:** Thank you for your suggestion. We are in the process of re-organizing this section and will add a note here when it is completed on our end.

---

> ### Author Response · Authors · 2024-11-22
> **Hope to Get Your Reply**
>
> Dear Reviewer Y9en,
>
> As the deadline is nearing, we wanted to gently follow up on our recent submission. Your feedback is highly valuable to us, and we would appreciate any updates or further guidance you might have regarding our revisions and responses.
>
> Thank you for your time and consideration.

---

> ### Author Response · Authors · 2024-11-25
> **Follow up update regarding w3**
>
> >**W3:** In fact, the explanations of conflict distribution and irrelevant prompts in the abstract and introduction are quite obscure and difficult to understand. I had to reread these sections several times, only gaining clarity after reading the methods section. This part may need reorganization.
> ---
>
> **Reply W3:** Following your suggestions, we have added two additional lines to clarify the explanation of the irrelevant prompts issue in the abstract (L16–L18) and introduction (L101–L107).
>
> We hope our explanations have clarified any unclear aspects of our proposed method. Your questions and feedback have been invaluable, and we sincerely appreciate the opportunity to address your concerns. If you feel that we have adequately resolved your queries and you are satisfied with the discussion, we would be grateful if you could consider revisiting the rating of our work.

---

> > ### Comment · Reviewer_Y9en · 2024-11-28
> >
> > Thank you for the author's response. I agree with the other reviewers' opinions. Although the author has explained it, the current motivation remains quite unclear. Including two more intuitive examples in Figure 1 would be helpful. I suggest the author refine the motivation further.

---

> ### Author Response · Authors · 2024-12-03
> **Wrapping up the discussion**
>
> We sincerely appreciate your time and thoughtful review!
>
> In our general comments, we provided both theoretical and experimental motivation for our work, along with a series of proofs demonstrating the effectiveness of DCPO compared to diffusion-DPO. As the rebuttal period approaches its conclusion, we would like to respectfully ask if you have any remaining concerns regarding the motivation behind our work, the effectiveness of our method (including the additional results provided), or any other aspects of our submission. We have been more than happy to address any remaining or new concerns during the discussion period.
>
> If our responses have adequately addressed your concerns, we kindly ask you to consider reevaluating our work in light of the updated information.
>
> Thank you once again for your valuable feedback!

---

### Official Review · Reviewer_CPj9 · 2024-10-31

**Soundness:** 3
**Presentation:** 2
**Contribution:** 2
**Rating:** 5
**Confidence:** 3

**Summary:**

The paper presents a preference optimization technique called Dual Caption Preference Optimization (DCPO). This method aims at improving text-to-image diffusion models. DCPO tackles issues inherent in current preference datasets, namely conflict distribution and irrelevant prompts, by introducing separate captions for preferred and less preferred images. This dual-caption approach is implemented through three methods: captioning, perturbation, and a hybrid method, all aimed at enhancing the clarity of distinctions between preferred and non-preferred images. Experimental results demonstrate that DCPO outperforms existing models across several benchmarks and metrics, including Stable Diffusion 2.1 and Diffusion-DPO.

**Strengths:**

1. The dual caption framework is reasonable. DCPO introduces a dual-caption system that effectively addresses the problem of overlapping distributions in existing datasets.
2. This paper achieves better performance. Demonstrated improvements across multiple metrics (e.g., Pickscore, CLIPscore) and benchmarks (e.g., GenEval) show that DCPO enhances image quality and relevance significantly.
3. The experimental results are analyzed in detail. The paper includes extensive quantitative and qualitative analysis, supporting the effectiveness of DCPO with various baselines and ablation studies.

**Weaknesses:**

1. The proposed method depends on the caption quality. The quality of generated captions significantly affects performance, and challenges remain in creating effective captions for less preferred images without straying out-of-distribution.
2. While DCPO demonstrates quantitative improvements across several metrics, the qualitative results (e.g., Figure 1) indicate that the visual distinctions between images generated by DCPO and baseline methods are not significant. This subtle difference may limit the perceived impact of DCPO in practical applications.
3. The DCPO has limited generalizability compared to real-world large-scale datasets. Although leveraging preferred and non-preferred images is a novel approach for enhancing diffusion models, high-quality, large-scale datasets from real-world settings often provide stronger improvements in model performance. This reliance on real-world data diminishes the relative advantage of DCPO, potentially limiting the distinctiveness of its contributions in scenarios where comprehensive datasets are available.
4. The LAION-2B and MSCOCO datasets are widely regarded benchmarks for image generation tasks, yet they are not discussed or evaluated within this study. The absence of experiments or comparisons involving LAION-2B raises questions about DCPO’s general applicability.

**Questions:**

Please address my concerns above.

---

> ### Author Response · Authors · 2024-11-19
> **Official Reply to Reviewer CPj9 part 1/2**
>
> We sincerely thank the reviewer for their insightful feedback on our paper. We look forward to engaging in a constructive discussion, as there is still plenty of time remaining in the discussion period. Below, we present a detailed response to your comments:
>
> ---
> > **W1:** The proposed method depends on the caption quality. The quality of generated captions significantly affects performance, and challenges remain in creating effective captions for less preferred images without straying out-of-distribution.
> ---
>
> **Reply to W1:** One of the key experiments in our paper investigates how the quality of generated captions influences the alignment performance of diffusion models. We first define high-quality captions as those with a higher correlation between image $ x $ and caption $ z $. In Table 5 in Appendix B, we reported the average CLIP score as a standard correlation metric.
>
> The results in Table 5 in Appendix B indicate that captions generated by LLaVA have better quality, evidenced by a higher average CLIP score compared to Emu2. Specifically, the difference in correlation scores between preferred and less preferred captions is approximately 4 for LLaVA and 3 for Emu2, both of which are larger than the original prompt (~2, as explained in Lines 340–357). However, despite the higher caption quality of LLaVA, the performance of DCPO-c-Emu2 remains comparable to DCPO-c-LLaVA, as shown in Tables 1 and 2.
>
> This suggests that while caption quality plays a role, it is not the primary determinant of performance. Instead, the critical factor is the difference in correlation between captions and the preferred and less preferred images. Evidence for this conclusion arises from the superior performance of DCPO-h, which achieves the largest correlation distance among DCPO-c and DCPO-p techniques, as shown in Figure 4. This supports the hypothesis that maximizing the correlation difference, rather than caption quality alone, is one of the keys to optimizing alignment performance.
>
> ---
> > **W2:** While DCPO demonstrates quantitative improvements across several metrics, the qualitative results (e.g., Figure 1) indicate that the visual distinctions between images generated by DCPO and baseline methods are not significant. This subtle difference may limit the perceived impact of DCPO in practical applications.
> ---
>
> **Reply to W2:** Thank you for this comment. We have updated Figure 1 in the revised version. Additionally, the reviewer can find more qualitative examples across different benchmarks in Appendix F.
>
> ---
> > **W3:** The DCPO has limited generalizability compared to real-world large-scale datasets. Although leveraging preferred and non-preferred images is a novel approach for enhancing diffusion models, high-quality, large-scale datasets from real-world settings often provide stronger improvements in model performance. This reliance on real-world data diminishes the relative advantage of DCPO, potentially limiting the distinctiveness of its contributions in scenarios where comprehensive datasets are available.
> ---
>
> **Reply to W3:** To the best of our knowledge, real-world datasets do not have a preference structure. Preference data requires selecting preferred and less preferred images from multiple images generated for the same prompt based on human judgment. While real-world datasets like MSCOCO and LAION-2B are valuable, they are not suitable for preference optimization due to the lack of this structure.
>
> Preference datasets are specifically used in post-training, a step following pre-training, to enhance the performance of pre-trained models, as outlined in Diffusion-DPO. For preference optimization, a dataset D ={$c, x^w, x^l$} is required, where $ x^w $ and $ x^l $ represent the preferred and less preferred images for the same prompt $ c $. This specialized data structure is essential for methods like DCPO to achieve their objectives.
>
>
> To demonstrate DCPO's generalizability, we fine-tuned Stable Diffusion 2.1 using Diffusion-DPO and DCPO on another high-quality preference dataset - Rapidata Image Generation Preference Dataset (RIGPD) [1]. The table below shows that DCPO variants consistently outperform Diffusion-DPO across multiple benchmarks, including Geneval, Pickscore, HPSv2.1, ImageReward, and CLIPscore.
>
> | Method (SD2.1)   | Geneval (Overall) | Pickscore | HPSv2.1 | ImageReward | CLIPscore |
> |-------------------|:-------------------:|:-----------:|:---------:|:-------------:|:-----------:|
> | Diffusion-DPO     | 0.4813            | 20.34     | 25.10   | 55.4        | 26.84     |
> | DCPO-c            | 0.4867            | **20.44**     | **25.43**   | **55.7**        | 26.86     |
> | DCPO-h            | **0.4978**   | 20.42     | 25.10   | 55.6| **26.91** |
>
> These results highlight DCPO's versatility and superior performance across benchmarks.
>
> **[1]** Rapidata Image Generation Preference Dataset (RIGPD): https://huggingface.co/datasets/Rapidata/700k_Human_Preference_Dataset_FLUX_SD3_MJ_DALLE3

---

> ### Author Response · Authors · 2024-11-19
> **Official Reply to Reviewer CPj9 part 2/2**
>
> ---
> > **W4:** The LAION-2B and MSCOCO datasets are widely regarded benchmarks for image generation tasks, yet they are not discussed or evaluated within this study. The absence of experiments or comparisons involving LAION-2B raises questions about DCPO’s general applicability.
> ---
>
> **Reply to W4:** LAION and MSCOCO are not equipped with preference data by design, and there is no publicly annotated version of them. Hence, we are unable to perform experiments on them. Our practice is well aligned with existing works such as [1], [2]. Still, for image generation purposes, we evaluated DCPO and Diffusion-DPO on the FID metric to provide a broader perspective on their performance, and DCPO consistently outperformed Diffusion-DPO.
>
> **[1]** Diffusion-DPO: https://arxiv.org/abs/2311.12908
>
> **[2]** MaPO: https://arxiv.org/abs/2406.06424

---

> ### Author Response · Authors · 2024-11-22
> **Hope to Get Your Reply**
>
> Dear Reviewer CPj9,
>
> As the deadline is nearing, we wanted to gently follow up on our recent submission. Your feedback is highly valuable to us, and we would appreciate any updates or further guidance you might have regarding our revisions and responses.
>
> Thank you for your time and consideration.

---

> > ### Comment · Reviewer_CPj9 · 2024-11-27
> >
> > Thank you for your response. Most of my concerns have been addressed. However, after reviewing all the reviewers' discussions, I still have concerns regarding the paper's motivation and novelty, so I will maintain my original score.

---

> > > ### Author Response · Authors · 2024-11-28
> > >
> > > Thank you for your response. We kindly ask the reviewer to clarify the concerns regarding motivation and novelty and highlight any similar work. We believe our paper's novelty is its key contribution, and we are actively conducting a new experiment to address any ambiguity. We also hope the reviewer will consider the extensive efforts and experiments we have already undertaken to address these concerns.

---

> ### Author Response · Authors · 2024-12-03
> **Wrapping up the discussion**
>
> We sincerely appreciate your time and thoughtful review!
>
> In our general comments, we provided both theoretical and experimental motivation for our work, along with a series of proofs demonstrating the effectiveness of DCPO compared to diffusion-DPO. As the rebuttal period approaches its conclusion, we would like to respectfully ask if you have any remaining concerns regarding the motivation behind our work, the effectiveness of our method (including the additional results provided), or any other aspects of our submission. We have been more than happy to address any remaining or new concerns during the discussion period.
>
> If our responses have adequately addressed your concerns, we kindly ask you to consider reevaluating our work in light of the updated information.
>
> Thank you once again for your valuable feedback!

---

### Official Review · Reviewer_f715 · 2024-11-03

**Soundness:** 2
**Presentation:** 3
**Contribution:** 2
**Rating:** 3
**Confidence:** 3

**Summary:**

This paper propose an improved method for aligning text to image diffusion models using human labeled preference datasets. Instead of using the original caption that is used to generate the preferred and less preferred image pair, the method propose to generate new captions from the generated images or original captions so as to increase/decrease the text image alignment for the generated images, which make the the distribution difference between preferred and less preferred data larger than using original captions.

Experiments using diffusion DPO method are conducted in various versions of caption image combinations, which show adding perturbed captions for less preferred image helps finetuned model get better performance on automatic metrics, including itemized metrics such as HPSv2, as well as side by side metrics using GPT-4o as judge.

**Strengths:**

The paper is well written with right amount of details in both main text and appendix. The proposed method is clear, and relativly straightforward to implement.

On a popular open source diffusion model ( SD 2.1), several experiments are done to ablate the design details of the proposed approach. The used set of metrics are comprehensive, including both single side evaluation such as HPSv2, as well as side by side evaluation such as the one using GPT4-o as judge.

**Weaknesses:**

The motivation behind the proposed approach is not clear to me.
For the conflict distribution challenge, when the distribution overlap becomes larger, the dataset is proposing a harder problem for the model to optimize, but it isn't necessary an issue as long as the two distributions are not identical. When the diffusion models's quality gets better, the two distribution will inevitably become more and more similar,  as both preferred and less preferred images from an optimized model will be closer to real human preference.  So it's more of the nature of the task itself, unless the task is defined differently.

From the description of L175-L180, the irrelevant problem is hardly a problem either. It is an inherently part of the objective in Eqn (1), where one way of minimizing Eqn(1) is to decrease $\log(p_{\theta}(x_{0:T}^l|z^l)$, which makes the model less likely to generate the less preferred image. So to me this is a desired behavior instead of a problem.

By changing the captions, authors changed a prefer/less prefer pair into two separate samples. In this sense it is no longer the original DPO problem, yet there is no clear connection between the original DPO formulation and the new problem e.g. is the new one an upper-bound of the original so minimizing the new problem potentially minimize the original one? or why solving the new problem will necessarily give better results than original DPO?

The change of captions made the problem closer to the KTO problem referenced in the paper, where text-image data are labeled by like and dislike binary labels. Please describe the connection and difference between the modified problem represented by the new data and the KTO problem formulation above.

It is great to conduct extensive experiments on SD 2.1, but the paper will be stronger if there are experiment results on other diffusion models, even if the experiments are not as complete as on SD 2.1.

**Questions:**

Despite the experiments suggests the proposed approach is better, it is unclear to me why this would be the case, any proofs or intuitions will help reader better understand it.

Several papers appear multiple times in the References section, please dedupe.

---

> ### Author Response · Authors · 2024-11-19
> **Official Reply to Reviewer f715 part 1/3**
>
> We sincerely thank the reviewer for their thoughtful feedback and the time dedicated to evaluating our paper. We hope our responses below address the concerns raised and support a potential reconsideration of the score.
>
> ---
> >**W1:** The motivation behind the proposed approach is not clear to me. For the conflict distribution challenge, when the distribution overlap becomes larger, the dataset is proposing a harder problem for the model to optimize, but it isn't necessary an issue as long as the two distributions are not identical.
> ---
> **Reply to W1:** Suppose we aim to optimize a policy model $ p_\theta $, such as a large language model by preference optimization methods. For this, we require a dataset $ D $ that includes both preferred ($ x^w $) and less preferred ($ x^l $) responses generated for the same prompt $ c $. The standard method involves generating multiple responses for a single input, ranking these responses through human evaluation or automated Judger models (e.g., GPT-4o), and selecting the best response as preferred and the least desirable as less preferred. This methodology is outlined in the OpenAI paper on Reinforcement Learning with Human Feedback (RLHF) [1].
>
> Also, in commonly generated preference datasets such as UltraFeedback-binarized [2][3] and UltraFeedback-PairRM [4][5], a similar approach is utilized with RLHF. However, we observed that existing vision-based preference datasets, such as Pick-a-Picv2, demonstrate a high correlation between preferred and less-preferred images—a phenomenon we refer to as "conflict distribution."
>
> Our hypothesis is that increasing the divergence between the correlation distributions of preferred and less preferred images can enhance the performance of diffusion models optimized using direct preference optimization methods. By addressing this "conflict distribution," our approach aims to make the optimization process more effective and improve overall model performance.
>
>
> **[1]** RLHF: https://arxiv.org/pdf/2203.02155
>
> **[2]** UltraFeedback Binarized: https://huggingface.co/datasets/HuggingFaceH4/ultrafeedback\_binarized
>
> **[3]** Zephyr: https://arxiv.org/abs/2310.16944
>
> **[4]** UltraFeedback PairRM: https://huggingface.co/datasets/princeton-nlp/llama3-ultrafeedback
>
> **[5]** SimPO: https://arxiv.org/abs/2405.14734
>
> ---
>
> >**W2:** When the diffusion models's quality gets better, the two distribution will inevitably become more and more similar, as both preferred and less preferred images from an optimized model will be closer to real human preference. So it's more of the nature of the task itself, unless the task is defined differently.
> ---
> **Reply to W2:** The goal of preference optimization is to train a diffusion model to distinguish between preferred and less preferred images generated for the same prompt, where the less preferred image exhibits some differences from the preferred image. Theoretically, if the preferred ($ x^w $) and less preferred ($ x^l $) images are too similar, the values of $ \log P(x^w|c) $ and $ \log P(x^l|c) $ will converge, leading to a loss value close to zero. This would hinder the model's ability to effectively learn the preferred distribution.
>
> Moreover, during optimization, the diffusion model is designed to increase the likelihood of $ x^w $ while decreasing the likelihood of $ x^l $. However, when there is a high degree of similarity between preferred and less preferred images, the model may become confused, struggling to differentiate between the two distributions effectively.
>
> To address the potential convergence of the preferred and less preferred distributions as models improve, we propose an alternative evaluation framework. Instead of assigning binary preference scores to image pairs (e.g., $ I_g = 1 $ and $ I_b = 0 $), we suggest adopting a soft-scoring system that captures varying degrees of alignment with human preference across multiple images. For example, in a set of four generated images, scores could indicate preference levels such as {$ I_1: 0.7, I_2: 0.5, I_3: 0.1, I_4: 0.6 $}, where $ I_1 $ represents the most preferred image and $ I_3 $ the least preferred.
>
> Using the FLUX model, we tested this framework on 100 samples, generating $ I_1 $ and $ I_2 $ from different seeds. The average difference across 100 samples was **0.4**, demonstrating that current models like FLUX still struggle to produce multiple well-aligned images from the same prompt, further supporting the relevance of our proposed framework.

---

> > ### Comment · Reviewer_f715 · 2024-11-26
> >
> > Really appreciate the replies to my questions. I want to get a clear story on the motivation and intuition of this work. So far it's still confusing. Let's keep on discussing.
> > Seems the argument on W1 and W2 are self contradictory. In reply to W1, the authors claim they observed high correlation between preferred and less-preferred images. In reply to W2 authors observed that advanced model FLUX struggle to produce multiple well-aligned images, and average difference is 0.4. Does this suggest the image correlations are not high? or the correlation is strong on worse models but weak on more recent almost SOTA models?

---

> > > ### Author Response · Authors · 2024-11-26
> > >
> > > Thank you for your response.
> > >
> > > In response to W1, we mention that *"Our hypothesis is that increasing the divergence between the **correlation distributions** of preferred and less preferred images can enhance the performance of diffusion models optimized using direct preference optimization methods. By addressing this "conflict distribution," our approach aims to make the optimization process more effective and improve overall model performance."* . The distribution we refer to here is that of the CLIPScore, which we refer to as the conflict distribution throughout our paper. To clarify, we refer to the distribution of the correlation measured by CLIPScore(s), which takes an input a text and an image, and returns their alignment as measured by CLIP. We find that this **correlation** is high between a {preferred image, caption} and {less-preferred image, caption} in existing preference datasets, i.e., the difference between the CLIPScore of {preferred image, caption} and {less-preferred image, caption} is similar.
> > >
> > > In response to W2, we argue that existing SOTA Models, such as FLUX, are also unable to generate multiple images aligned with the prompt. This was in response to the question that despite advancements in T2I generation, there is still a substantial gap in generating well-aligned images (as denoted by the **0.4** difference in our results).
> > >
> > > We hope this clarifies any confusion. Please let us know if you have more questions. Thank you.

---

> ### Author Response · Authors · 2024-11-19
> **Official Reply to Reviewer f715 part 2/3**
>
> >**W3:** From the description of L175-L180, the irrelevant problem is hardly a problem either. It is an inherently part of the objective in Eqn (1), where one way of minimizing Eqn(1) is to decrease $log(p(xi|zl))$, which makes the model less likely to generate the less preferred image. So to me this is a desired behavior instead of a problem.
>
> ---
> **Reply to W3:** We observed that the prompt $ c $ often does not serve as a meaningful caption for less preferred images (see Figure 2 in the main text and Figure 10 in Appendix C) because it may contain irrelevant information. We refer to this as the "irrelevant prompt" problem. The key question is: **Why is an irrelevant prompt problematic?**
>
> U-Net predicts the added noise for both preferred and less preferred images conditioned on a prompt, as shown in Equation 2. If the prompt is irrelevant for the less preferred images, U-Net’s ability to make accurate predictions becomes limited. This issue is supported by findings in the DALL-E 3 paper [1], which demonstrates that more relevant captions improve U-Net’s performance in predicting noise. Building on this, we generated relevant captions for the less preferred images using a captioning method that conditions the original prompt.
>
> During direct preference optimization, $ \log P(x^w|c) $ increases while $ \log P(x^l|c) $ decreases, where $ x^w $ and $ x^l $ represent the preferred and less preferred images generated for the same prompt $ c $. Theoretically, if a prompt $ c_1 $ has a stronger correlation with both the preferred and less preferred images, the optimization process is more effective, resulting in better model performance. Intuitively, a diffusion model better understands why humans prefer image $ x^w $. Because the prompt $ c_1 $ contains more relevant information. Thus, prompts with high correlation between $ x^w $ and $ x^l $ can influence $ \log P(x|c) $ more effectively during optimization. The results in Table 4 provide strong evidence supporting this explanation.
>
> Inspired by this behavior, we hypothesized that generating a more suitable caption for less preferred images, distinct from the original prompt, helps decrease the likelihood of less preferred images during optimization. This aligns with our findings and further validates the importance of addressing the irrelevant prompt issue.
>
> **[1]** DALL-E 3: https://cdn.openai.com/papers/dall-e-3.pdf
>
> ---
> >**W4:** By changing the captions, authors changed a prefer/less prefer pair into two separate samples. In this sense it is no longer the original DPO problem, yet there is no clear connection between the original DPO formulation and the new problem e.g. is the new one an upper-bound of the original so minimizing the new problem potentially minimize the original one? or why solving the new problem will necessarily give better results than original DPO?
> ---
>
> **Reply to W4:** In the DCPO framework, we modify the original prompt by replacing it with a caption generated by a captioning model $ Q_\phi $. This modification enables better alignment with the less preferred images while maintaining the DCPO in the original task space.
>
> To establish the equivalence with the original Diffusion-DPO problem, we theoretically demonstrate that Diffusion-DPO provides a lower bound of the DCPO loss function. In the DCPO framework, the caption $ z $ is generated by the captioning model $ Q_\phi $ conditioned on the image $ x $ and the original prompt $ c $, i.e., $ z \sim Q_\phi(z \mid x, c) $. Importantly, in scenarios where $ Q_\phi $ generates a caption $ z $ that closely matches the original prompt $ c $, we effectively have $ z \simeq c $. In this case, substituting $ z $ for $ c $ in the DCPO loss function produces the original Diffusion-DPO loss function (refer to Equation 9 in Appendix A).
>
> This theoretical equivalence ensures that minimizing the DCPO loss function aligns with minimizing the Diffusion-DPO loss. Moreover, the modification introduced by $ Q_\phi $ helps address challenges arising from irrelevant prompts in the original framework, which we have shown to improve alignment and performance on various benchmarks. Thus, solving the DCPO problem not only retains the original Diffusion-DPO framework’s objectives but also provides additional flexibility for addressing inherent limitations, leading to better results.

---

> ### Author Response · Authors · 2024-11-19
> **Official Reply to Reviewer f715 part 3/3**
>
> >**W5** The change of captions made the problem closer to the KTO problem referenced in the paper, where text-image data are labeled by like and dislike binary labels. Please describe the connection and difference between the modified problem represented by the new data and the KTO problem formulation above.
>
> ---
> **Reply to W5:** Assume a preference dataset D ={$c, x^w, x^l$}, where $ x^w $ and $ x^l $ represent the preferred and less preferred images for the prompt $ c $. The Diffusion-KTO [1] hypothesizes optimizing a diffusion model using only a single preference label based on whether an image $ x $ is suitable or unsuitable for a given prompt $ c $. Thus, Diffusion-KTO utilizes a dataset D ={$c, x$}, where $ x $ is a generated image corresponding to prompt $ c $.
>
> This hypothesis is fundamentally different from ours. While Diffusion-KTO focuses on binary preferences (like/dislike) for individual image-prompt pairs, our approach involves paired preferences. We do not claim that having two preferences is problematic; rather, we observed that using the same prompt $ c $ for both preferred and less preferred images may not be ideal. To address this, we propose optimizing a diffusion model using a dataset D ={$ z^w, z^l, x^w, x^l$}, where $ z^w $ and $ z^l $ are captions generated by a captioning model $ Q_\phi $ for the preferred and less preferred images with respect to the original prompt, respectively.
>
> However, we compared Diffusion-KTO on various benchmarks, and the results are presented in the following table. This comparison highlights how the differences in problem formulation influence the model's performance and further emphasizes the distinct nature of our proposed approach relative to Diffusion-KTO.
>
> | Method (SD2.1)   | Geneval (Overall) | Pickscore | HPSv2.1 | ImageReward | CLIPscore |
> |---------|:----------:|:-------:|:------:|:-------:|:--------:|
> | Diffusion-DPO | 0.4857   | 20.36     | 25.10   | 56.4 | 26.98 |
> | Diffusion-KTO  | 0.5008     | 20.41     | 24.80   | 55.5  | 26.95  |
> | DCPO-h  | **0.5100**   | **20.57** | **25.62** | **58.2**  | **27.13** |
>
>
> **[1]** Diffusion-KTO: https://arxiv.org/abs/2404.04465
>
> ---
> >**W6:** It is great to conduct extensive experiments on SD 2.1, but the paper will be stronger if there are experiment results on other diffusion models, even if the experiments are not as complete as on SD 2.1.
> ---
> **Reply to W6:** In response to the reviewer's suggestion, we conducted additional experiments to evaluate the performance of DCPO on the SDXL model. Because SDXL is a large model, we perform LoRA fine-tuning with minimal hyper-parameter search.
>
> The results presented in the following table demonstrate that DCPO outperforms Diffusion-DPO on metrics such as Pickscore, Geneval, HPSv2, and CLIPscore. Additionally, DCPO achieves performance comparable to Diffusion-DPO on ImageReward. These findings highlight the generalizability and effectiveness of DCPO across different diffusion models.
>
> | Method (SDXL)       | Geneval (Overall) | Pickscore   | HPSv2.1   | ImageReward | CLIPscore   |
> |-------------|:-------------------:|:-----------:|:---------:|:------------:|:-----------:|
> | Diffusion-DPO | 0.5645     | 21.77       | 28.64     | 71.2 | 28.61       |
> | DCPO-c  | **0.5758**   | **21.87**   | **28.65** | 71.2 | **28.63**   |
> | DCPO-h-weak| 0.5704   | **21.87**   | 28.64     | 71.2  | 28.62       |
> | DCPO-h-medium | 0.5700  | 21.86  | 28.64| 71.2   | **28.63**   |
> | DCPO-h-strong| 0.5696 | 21.86  | 28.64  | 71.2   | 28.62  |
>
> ---
> >**Q1:** Despite the experiments suggests the proposed approach is better, it is unclear to me why this would be the case, any proofs or intuitions will help reader better understand it.
> ---
>
> **Reply to Q1:** The Diffusion-DPO objective is designed for preference optimization using a dataset D ={$c, x^w, x^l$}, where $ x^w $ and $ x^l $ represent preferred and less preferred images for a given prompt $ c $. DCPO enhances this framework by replacing the prompt $ c $ with a caption $ z $, generated by a captioning model $ Q_\phi $, which provides more relevant and contextually aligned information about each image. This adjustment improves performance by enabling better contextual conditioning, where $ z^l $ captures specific details relevant to $ x^l $ that $ c $ might lack. It also reduces noise by eliminating irrelevant details in $ c $, supporting the model in distinguishing $ x^w $ from $ x^l $ more effectively. Additionally, conditioning on $ z^l $ simplifies dependencies, ensuring the model focuses entirely on the necessary information for preference evaluation. These improvements result in better preference optimization, as demonstrated by the stronger correlation between captions and their corresponding images compared to the original prompts.
>
> ---
> >**Q2:** Several papers appear multiple times in the References section, please dedupe.
> ---
>
> **Reply to Q2:** Thank you for pointing this out! We’ve fixed the issue in the updated version.

---

> ### Author Response · Authors · 2024-11-22
> **Hope to Get Your Reply**
>
> Dear Reviewer f715,
>
> As the deadline is nearing, we wanted to gently follow up on our recent submission. Your feedback is highly valuable to us, and we would appreciate any updates or further guidance you might have regarding our revisions and responses.
>
> Thank you for your time and consideration.

---

> ### Author Response · Authors · 2024-12-03
> **Wrapping up the discussion**
>
> We sincerely appreciate your valuable time and thoughtful review.
>
> Regarding your concern about using different captions for the preferred and less preferred images, we would like to clarify that our approach aligns with prior work. As a side note, we have found that a recent paper accepted at ICML proposes DOVE, a loss function in a design similar to our DCPO, to address the limitations of the DPO loss function in language models. Similarly to our method, DOVE generates distinct instruction inputs for the less preferred examples, resulting in different inputs for both the preferred and the less preferred cases. We kindly refer you to [1], and its Figure 1 and Equation 2, for reference.
>
> As the rebuttal period is near its conclusion, we respectfully ask if you have any further concerns regarding the motivation behind our work, the effectiveness of our method (including the additional results provided) or any other aspects of our submission. We have been more than happy to address any remaining questions or new ones during the remaining discussion period.
>
> Thank you again for your constructive feedback!
>
> [1] DOVE: https://openreview.net/pdf?id=AzMnkF0jRT

---

### Official Review · Reviewer_tyap · 2024-11-04

**Soundness:** 3
**Presentation:** 3
**Contribution:** 3
**Rating:** 5
**Confidence:** 3

**Summary:**

This work first presents the conflict distribution issue in preference datasets, where preferred and less-preferred images generated from the same project exhibit significant overlap. For this issue, they introduce the Captioning and Perturbation methods: generate a caption based on the image and the prompt, create three levels of perturbation from the prompt. They also explore the irrelevant prompt issue in previous DPO methods and propose Dual Caption Preference Optimization (DCPO) to improve diffusion model alignment. Lastly, they show promising results compared to the existing methods.

**Strengths:**

1. The paper is well-organized and easy to follow. Figures are clear to read, such as Figure 2.
2. The story is complete: they propose hypothesis and then use experimental results to verify them in Sec 3.3 with clear ablation studies.
3. The problem setup is clear. They also provide enough details to reproduce the work.

**Weaknesses:**

1. My biggest concern is about the generalization of the approach method in the development of diffusion models. For example, in Figure 2, it is easy to distinguish the preferred and less-preferred image as the latter one even does not align with the original prompt. What if the model's development is already beyond the alignment stage? The current positive/negative samples are only about alignment, what about more advanced difference if both have enough alignment?

2. Line 188-189, could you explain more details on how to get the preferred and less-preferred images? Human annotation?

3. It would be beneficial to highlight the difference between medium and strong permutation. Do we have a way to quantify the difference between them? Are they controllable generated? Why do we need medium permutations? Would weak/strong be enough?

4. In terms of GPT-4o evaluation, does it matter for showing the images together or showing them separately? And how about the order of showing them to GOT-4o if showing separately?

**Questions:**

See weaknesses.

---

> ### Author Response · Authors · 2024-11-19
> **Official Reply to Reviewer tyap part 1/2**
>
> Thank you for your valuable feedback. We have thoughtfully addressed your comments below and look forward to discussing any remaining concerns during the discussion period to facilitate a positive re-evaluation of the score.
>
> ---
> > **W1:** My biggest concern is about the generalization of the approach method in the development of diffusion models. For example, in Figure 2, it is easy to distinguish the preferred and less-preferred image as the latter one even does not align with the original prompt. What if the model's development is already beyond the alignment stage? The current positive/negative samples are only about alignment, what about more advanced difference if both have enough alignment?
> ---
> **Reply to W1:** Thank you for your question. In this study, we follow the prevailing paradigm in preference data, where for a given text prompt $P$, an image $I_g$ is deemed preferred, while an image $I_b$ is less favored. We first observe that even leading models, such as FLUX, currently struggle to generate multiple aligned images from the same prompt. This current limitation highlights the significant potential for improvement in producing images with robust alignment.
>
> Assuming future advancements close this alignment gap, we propose an alternative evaluation framework: rather than assigning binary preference scores to image pairs (i.e., $I_g = 1$ and  $I_b = 0 $), we envision a soft-scoring system that captures varying degrees of alignment with human preference across multiple images. For instance, if four images are generated, they could be ranked with scores indicating preference levels, such as  {$ I_1 : 0.7, I_2 : 0.5, I_3 : 0.1, I_4 : 0.6 $}, where $ I_1$ is the most preferred and $I_3$ the least preferred.
>
> To test this approach, we generated 100 samples $I_1$ and $I_2$ using different seeds with FLUX and asked two volunteers to assign soft scores. The observed average difference of **0.4** demonstrates that current models like FLUX face challenges in producing multiple well-aligned images from the same prompt, emphasizing the importance of our proposed framework.
>
> ---
> >**W2:** Line 188-189, could you explain more details on how to get the preferred and less-preferred images? Human annotation?
> ---
> **Reply to W2:** In our experiments, we use the Pick-a-Pic v2 dataset, which is human annotated. This dataset is constructed where, for a given prompt, 2 images are generated; the user then chooses the preferred image between the two.
>
> - For more details, please refer to https://stability.ai/research/pick-a-pic.
>
> ---
> > **W3:** In terms of GPT-4o evaluation, does it matter for showing the images together or showing them separately? And how about the order of showing them to GPT-4o if showing separately?
> ---
>
> **Reply to W3:** We follow standard practice, as in Diffusion-DPO [1] and MaPO [2], by showing two images side by side for comparison. To address positional bias in GPT-4o's evaluations, we alternate the positions of the images across different criteria (explained in Section 3).
>
> The results below show that DCPO consistently achieves better performance than Diffusion-DPO, even when positional bias is accounted for.
>
> | Model             | General Preference (Win Rate%) | Visual Appeal (Win Rate%) | Prompt Alignment (Win Rate%) |
> |--------------------|:--------------------:|:---------------:|:------------------:|
> | SD2.1-DCPO-h      | **58%**            | **64.5%**     |    **56.5%**        |
> | SD2.1-DPO         | 42%                | 35.5%         | 43.5%            |
>
> This demonstrates that DCPO provides more reliable results under unbiased conditions.

---

> ### Author Response · Authors · 2024-11-19
> **Official Reply to Reviewer tyap part 2/2**
>
> > **W4:** It would be beneficial to highlight the difference between medium and strong permutation. Do we have a way to quantify the difference between them? Are they controllable generated? Why do we need medium permutations? Would weak/strong be enough?
> ---
>
> **Reply to W4:**  In our study, perturbations are controllably generated using DIPPER, a model that allows configuration via a 'lexicon diversity' parameter, which ranges from 1 to 100. Lexicon diversity measures the likelihood of a perturbation introducing alternative expressions or synonyms. By setting this parameter to 40, 60, and 80, we create weak, medium, and strong perturbations, respectively.
>
> Qualitatively, weak perturbations replace certain words in the original caption with synonyms, medium perturbations combine synonym swapping with the reordering of words, and strong perturbations paraphrase the caption into entirely different sentence structures.
>
> Quantitatively, Figure 4 illustrates the differences across these levels. Based on the CLIPScore distribution of the original captions (turquoise histogram in (a)), we observe that stronger perturbations shift the CLIPScores further left. Medium perturbations (c) remain closer to the original captions in terms of mean and standard deviation compared to strong perturbations (d), demonstrating that medium perturbations achieve a balance between preserving the original meaning and introducing diversity.
>
> ---
> **[1]** Diffusion-DPO: https://arxiv.org/abs/2311.12908
>
> **[2]** MaPO: https://arxiv.org/abs/2406.06424

---

> ### Author Response · Authors · 2024-11-22
> **Hope to Get Your Reply**
>
> Dear Reviewer tyap,
>
> As the deadline is nearing, we wanted to gently follow up on our recent submission. Your feedback is highly valuable to us, and we would appreciate any updates or further guidance you might have regarding our revisions and responses.
>
> Thank you for your time and consideration.

---

> > ### Comment · Reviewer_tyap · 2024-11-26
> > **Thank you for your response.**
> >
> > Thank you for your replies to my questions. I have read all reviews and discussions from reviewers and authors. I agree with Reviewer f715 and am also getting confused about the motivation. Therefore, I may tend to downgrade my rate if this not cleared.
> >
> > >I want to get a clear story on the motivation and intuition of this work. So far it's still confusing. Let's keep on discussing. Seems the argument on W1 and W2 are self contradictory. In reply to W1, the authors claim they observed high correlation between preferred and less-preferred images. In reply to W2 authors observed that advanced model FLUX struggle to produce multiple well-aligned images, and average difference is 0.4. Does this suggest the image correlations are not high? or the correlation is strong on worse models but weak on more recent almost SOTA models?

---

> > > ### Author Response · Authors · 2024-11-26
> > >
> > > Thank you for your response.
> > >
> > > In response to W1, we mention that *"Our hypothesis is that increasing the divergence between the **correlation distributions** of preferred and less preferred images can enhance the performance of diffusion models optimized using direct preference optimization methods. By addressing this "conflict distribution," our approach aims to make the optimization process more effective and improve overall model performance."* . The distribution we refer to here is that of the CLIPScore, which we refer to as the conflict distribution throughout our paper. To clarify, we refer to the distribution of the correlation measured by CLIPScore(s), which takes an input a text and an image, and returns their alignment as measured by CLIP. We find that this **correlation** is high between a {preferred image, caption} and {less-preferred image, caption} in existing preference datasets, i.e., the difference between the CLIPScore of {preferred image, caption} and {less-preferred image, caption} is similar.
> > >
> > > In response to W2, we argue that existing SOTA Models, such as FLUX, are also unable to generate multiple images aligned with the prompt. This was in response to the question that despite advancements in T2I generation, there is still a substantial gap in generating well-aligned images (as denoted by the **0.4** difference in our results).
> > >
> > > We hope this clarifies any confusion. Please let us know if you have more questions. Thank you.

---

> > > ### Author Response · Authors · 2024-12-03
> > > **Wrapping up the discussion**
> > >
> > > We sincerely appreciate your time and thoughtful review!
> > >
> > > In our general comments, we provided both theoretical and experimental motivation for our work, along with a series of proofs demonstrating the effectiveness of DCPO compared to diffusion-DPO. As the rebuttal period approaches its conclusion, we would like to respectfully ask if you have any remaining concerns regarding the motivation behind our work, the effectiveness of our method (including the additional results provided), or any other aspects of our submission. We have been more than happy to address any remaining or new concerns during the discussion period.
> > >
> > > If our responses have adequately addressed your concerns, we kindly ask you to consider reevaluating our work in light of the updated information.
> > >
> > > Thank you once again for your valuable feedback!

---

### Author Response · Authors · 2024-11-28
**Global comment to the reviewers**

Thank you to the reviewers for their comments and feedback.

---
- We found a recent COLM 2024 paper in which the authors, in the future work section, suggested aligning diffusion models using two distinct captions and revising the DPO loss function. In DCPO, the captioning model functions similarly to the Refiner described in their work. Further details are in Appendix D of the paper:

[1] From r to Q∗: Your Language Model is Secretly a Q\-Function: https://arxiv.org/abs/2404.12358

---
**Our motivations follow the same spirit and we elaborate them as follows.** In preference datasets, preferred and less preferred images must exhibit clear differences. If these differences are minimal, the DPO loss function collapses theoretically because $\log P(x_w|c) \simeq \log P(x_l|c) where:
x_w \simeq x_l$, reducing the loss to near zero. To address this, current preference datasets select different images as preferred and less preferred for a given prompt $c$ (addressing W1 and W2 for reviewer f715).

An unexplored area in preference optimization for diffusion models is the effect of improving captions and using distinct captions. We approached this by enhancing the original captions through captioning models. As shown in Table 4, improved captions enhance the performance of preference optimization. The reasoning is theoretical: when a caption $c_1$ has higher semantic similarity with the image $x$ than $c_2$, $\log P(x_w|c)$ in the first term of the DPO loss improves, leading to better alignment performance. Thus, improving captions boosts DPO performance by improving $ \log P(x_w|c)$.

- For more details, please refer to the original DPO paper: https://arxiv.org/abs/2305.18290

**Hypothesis 1:** If a distinct caption for the less preferred image correctly describes it, this will improve $\log P(x_l|c)$ in the DPO loss function.

To test this hypothesis, we conducted the following two Experiments:

**Experiment 1:** Optimize a diffusion model with DPO using irrelevant captions for less preferred images created by heavily perturbing the prompt.

**Experiment 2:** Generate captions for less preferred images using a captioning model based on the original caption.

| Method (SD2.1) |(Caption of Preferred, Less Preferred Images)| GenEval (Overall) | Pickscore | HPSv2.1 | ImageReward | CLIPscore |
|-|-|-|-|-|-|-|
| DPO | original caption c for both| 0.4857| 20.36| 25.10| 56.4| 26.98|
| DPO (**Experiment 1**)| original caption c, perturbed original caption| 0.4852| 20.21| 25.34| 53.1| 26.87|
| DPO (**Experiment 2**)| original caption c, generated caption for less preferred| 0.4870| 20.41| 25.11| 56.5| 26.98|

As shown in the above table, alignment performance improves when aligned captions are used for less preferred images, supporting Hypothesis 1. This resolves the 'irrelevant prompts' issue in both the optimization process and the datasets.

**Observation 1:** While alignment performance increases by generating a good caption for less preferred images, the improvement is modest. Our data analysis reveals that the CLIPscores of preferred and less preferred images are often similar (see Figure 3). The similarity arises because the caption for the less preferred image was generated by a captioning model, leading to minimal differences.

**Hypothesis 2:** Based on observation 1, we hypothesize that increasing the difference in average CLIPscore between preferred and less preferred images will improve the alignment performance of the diffusion model.

To test this, we generated two distinct captions for preferred and less preferred images, respectively. The goal was to improve $ \log P(x_w|c) $ and $\log P(x_l|c)$ in the DPO loss function by having correspondingly more suitable captions. The results in Tables 1 and 2 confirm that DCPO significantly improves alignment performance across different benchmarks.

---
**For Reviewer Y9en - Difference between Conflict Distribution and Irrelevant Prompts** While both issues share similarities, we argue they are not exact duplicates according to the two Experiments we mentioned earlier. In Experiment 1, the difference between the CLIPscore of the preferred image with the original caption and the less preferred image with the perturbed caption is large, meaning there is no conflict distribution. It shows that having little conflict distribution in fact indicates to alignment performance decrease due to irrelevant prompts. In Experiment 2, the alignment performance is now better than the original DPO, but having two similar distributions of CLIPscore between preferred and less preferred images restricts the performance from improving futhermore. Thus, while Conflict Distribution and Irrelevant Prompts overlap in certain aspects, they are in fact distinct problems.

---
We thank all reviewers for their constructive suggestions. We hope the reviewers take into account the comprehensive analyses and experiments conducted during the rebuttal and reconsider their scores.

---

> ### Comment · Reviewer_f715 · 2024-11-29
>
> Really appreciate authors' efforts on the clarifications.
>
> Hypothesis 1: not sure what does "improve" mean. I can see that a more relevant caption $z_l$ will increase $\logP(x_l|c)$ to $\logP(x_l|z_l)$ which increase the loss. So it to some degree sets an upper bound to the original loss, but it is unclear why optimizing this upper bound would be necessarily better than optimizing the original loss.
>
> Overall I'd hold a neutral position if this paper simply states that through experiments they found generating $z_l$ and $z_w$ , and use them in DPO instead of $c$ yields better performance. It's quite unfortunate that so far none of the motivations sounds reasonable to me. It seems other reviewers are also confused.

---

> > ### Author Response · Authors · 2024-12-01
> > **Formal Proofs on DCPO's Motivation and Methodology - Part 1**
> >
> > We thank the reviewer for responding to our comment. We would like to further clarify that the Diffusion-DPO loss function does not include $\log P(x^l | c)$  in the final loss function. In the Diffusion-DPO paper [1], the authors have stated that incorporating $\log P(x^w | c)$ and $\log P(x^l | c)$ into the Diffusion-DPO loss function is *intractable and inefficient*, as we have reiterated in L205-206. For more details on Diffusion-DPO, we refer the reviewers to Section 4 of the Diffusion-DPO paper. In short, the final form of the Diffusion-DPO loss is similar to our DCPO loss, as shown in L211-214.
> >
> > [1] Diffusion-DPO: https://arxiv.org/abs/2311.12908
> >
> > ---
> > The Diffusion-DPO loss $L(\theta)$ can be expressed as the following equation:
> > $$L(\theta) = -\mathbb{E}_{(\mathbf{x}_0^w, \mathbf{x}_0^l, \mathbf{c}) \sim \mathcal{D}, \, t \sim \mathcal{U}(0, T), \, \mathbf{x}_t^w \sim q(\mathbf{x}_t^w | \mathbf{x}_0^w), \, \mathbf{x}_t^l \sim q(\mathbf{x}_t^l | \mathbf{x}_0^l)}
> > $$
> >
> > $$\Big[\log \sigma ( -\beta T \omega(\lambda_t) \cdot ( \Delta_{preferred} - \Delta_{less-preferred} ) ) \Big]
> > $$
> >
> > where, in actuality,
> >
> > $$ \Delta_{\text{preferred}} = \| \boldsymbol{\epsilon}^w - \boldsymbol{\epsilon}_\theta(\mathbf{x}_t^w, t, \mathbf{c}) \|_2^2 - $$
> >
> > $$\| \boldsymbol{\epsilon}^w - \boldsymbol{\epsilon}_{\text{ref}}(\mathbf{x}_t^w, t, \mathbf{c}) \|_2^2$$
> >
> > and
> >
> > $$
> > \Delta_{\text{less-preferred}} = \| \boldsymbol{\epsilon}^l - \boldsymbol{\epsilon}_\theta(\mathbf{x}_t^l, t, \mathbf{c}) \|_2^2 - $$
> >
> > $$\| \boldsymbol{\epsilon}^l - \boldsymbol{\epsilon}_{\text{ref}}(\mathbf{x}_t^l, t, \mathbf{c}) \|_2^2$$
> >
> > ---
> > **Our Motivation and Methodology.** We are motivated to demonstrate that, by having a larger difference between $\Delta_{\text{less-preferred}}$ and $\Delta_{\text{preferred}}$, a much clearer distinction between preferred and less preferred image-caption pairs contributes to more optimized $L(\theta)$ in terms of soft-margin optimization.
> >
> > Our DCPO paradigm enforces such larger difference between $\Delta_{\text{less-preferred}}$ and $\Delta_{\text{preferred}}$ on the following two factors. On $\Delta_{\text{less-preferred}}$, we replace the original shared caption, a.k.a. the prompt $c$ with a generated caption $z^l$ more suitable for the less-preferred image $x^l$ using $Q(z^l|x^l, c)$, in order to decrease $\Delta_{\text{less-preferred}}$. Likewise, on $\Delta_{\text{preferred}}$, we replace $c$ with an independently generated caption $z^w$ for the preferred image $x^w$ using $Q(z^w|x^w, c)$, in order to increase $\Delta_{\text{preferred}}$.
> >
> > In the following sections, we present the formal proofs on why our methodology leads to a more optimized $L(\theta)$ of a Diffusion-based model and, consequently, better performance in preference alignment tasks. The proofs will be incorporated into the camera-ready version of the paper.
> >
> > ---
> > **Proof 1: Increasing the difference between $\Delta_{\text{preferred}}$ and $ \Delta_{\text{less-preferred}}$ improves the optimization of $L(\theta)$.**
> >
> > For better clarity, the loss function $L(\theta)$ can be written as:
> > $$
> > L(\theta) = -\mathbb{E} \left[ \log \sigma \big( -\beta T \omega(\lambda_t) \cdot M \big) \right]
> > $$
> >
> > where $\sigma(x) = \frac{1}{1 + e^{-x}} $, a.k.a. the sigmoid function that squashes its input $ x $ into the range $ (0, 1) $ and $\ M = \Delta_{\text{preferred}} - \Delta_{\text{less-preferred}} $, a.k.a. the margin between the respective importance of the preferred and less preferred predictions.
> >
> > Characteristically, the gradient of $ \sigma(x) $ is at its maximum near $ x = 0 $ and decreases as $ |x| $ increases.
> > A larger margin in terms of $M$ makes it easier for the optimization to drive the sigmoid function towards its asymptotes, reducing loss.
> >
> > - When $ M $ is small ($ |M| \approx 0 $): The sigmoid $ \sigma(-\beta T \omega(\lambda_t) \cdot M) $ is near 0.5 (its midpoint). Also, the gradient of $ \log \sigma(x) $ is the largest near this point, meaning the model struggles to differentiate between preferred and less preferred predictions effectively.
> >
> > - When $ M $ is large ($ |M| \gg 0 $): The sigmoid $ \sigma(-\beta T \omega(\lambda_t) \cdot M) $ moves closer to 0 or 1, depending on the sign of $ M $. For a well-aligned model, if the preferred predictions are correct, $ M > 0 $ and $ \sigma(-\beta T \omega(\lambda_t) \cdot M) $ approach 1, thus minimizing the loss.
> >
> > Intuitively, an ideally large $ M $ represents a clear distinction between the preferred image-caption versus the less preferred image-caption. Thus, by maximizing $ M $, we may push the loss $ L(\theta) $ towards its minimum, leading to better soft-margin optimization.

---

> > > ### Comment · Reviewer_f715 · 2024-12-01
> > >
> > > I would like to express my gratitude towards authors again for clarifying the statements.
> > >
> > > The term $\log P(x_l|c)$ is used simply by following the authors' original response stated as hypothesis 1, where  $\log P(x_l|c)$  is one term in the loss function. This is not in conflict with the fact that eventually this term is reduced to L2 loss on noise prediction and ground-truth noise.
> > >
> > > In response to Proof 1, I agree that bigger $M$ does lower the loss, but it simply means $L(\theta|z_l, z_w) < L(\theta| c)$ if indeed $M$ using $(z_l, z_w)$ is bigger than using $c$. However, I would stick to my opinion that I am not sure what aspect of the optimization is "improved".
> > >
> > > To illustrate my point, let's assume a loss function family of $L(\theta| a, b, c) = a \theta^2 + b \theta + c$, by changing $a, b, c$, we can have two losses $L_1(\theta) = \theta^2$,  $L_2(\theta) = 0.5(\theta-1)^2- 1$, and we can show $L_2 \leq L_1$. Then I don't see why would I say optimization of $L_2$ is improved compared to $L_1$, as they are two problems, and the optimizer $\theta^*_2$ isn't necessarily equal to $\theta^*_1$, nor does performance of model $\theta^*2$ has to be better than $\theta^*_1$ since we didn't even introduce the original task at all.
> > >
> > > Back to the original paper's setting, the DCPO simply setup another loss function different from Diffusion-DPO,  and it is possible that this new loss is lower than the original loss for any $\theta$. But just as the $L_1$ and $L_2$ example above,  I don't know why optimizing the DCPO loss would necessarily improve the optimization problem in theory, or lead to "better performance in preference alignment tasks".
> > >
> > > We may argue that through experiments, we see that optimizing the new loss gives better results in terms of various evaluation metrics. But still this as a result doesn't prove the motivation is sound in theory.

---

> > > > ### Author Response · Authors · 2024-12-01
> > > >
> > > > We appreciate the reviewer's feedback and the acknowledgment that optimizing a diffusion model with the DCPO loss function decreases the loss. However, we are somewhat confused by the contradictory comments made throughout the review process.
> > > >
> > > > In the original review, the reviewer stated:
> > > > >"From the description of L175-L180, the irrelevant problem is hardly a problem either. It is inherently part of the objective in Eqn (1), where one way of minimizing Eqn (1) is to decrease $\log P_\theta(x^l_{0:T}|z^l)$, which makes the model less likely to generate the less preferred image. So to me, this is a desired behavior instead of a problem."
> > > >
> > > > In the reply comment, the reviewer reiterated:
> > > > >"I can see that a more relevant caption will increase $\log P(x_l|c)$ to $\log P(x^l|z^l)$, which increases the loss."
> > > >
> > > > However, in the final reply, the reviewer also explicitly stated:
> > > > >"In response to Proof 1, I agree that a larger $M$ does lower the loss, but it simply means $L(\theta|z_l, z_w) < L(\theta|c)$ if indeed $M$ using $(z^w, z^l)$ is larger than when using $c$."
> > > >
> > > > These comments seem inconsistent, and we would like to ask for clarification. Specifically, does optimizing a diffusion model with DCPO increase or decrease the loss?
> > > >
> > > > Furthermore, the 'loss family' example provided by the reviewer seems unclear to us. After all, Diffusion-DPO and DCPO share the same primary objective: to obtain a diffusion model that achieves better alignment with the given prompt, the less-preferred image, and the preferred image. Our DCPO methodology indirectly uses the prompt by instead using two respectively generated captions for the two images without additional fine-tuning or training on image captioning whatsoever. If the reviewer's concern relates to the validity of the Markov Decision Process (MDP) formulation of diffusion in the DCPO framework, we refer to [1], which demonstrates that conditioning the denoising process with $z^w$ and $z^l$ remains a valid MDP for diffusion.
> > > >
> > > > When $\Delta_{\text{preferred}} - \Delta_{\text{less-preferred}}$ increases as shown in Proof 1, the loss function decreases, indicating that the neural network $\epsilon_\theta $ is improving its ability to predict the added noise
> > > > $\boldsymbol{\epsilon} $ during the denoising process. This improvement is critical because accurate noise prediction ensures that each step of the denoising process moves the noisy image closer to its clean form, resulting in higher-quality, more realistic image generation. A lower loss also sharpens the alignment between the model's output and the conditioning input (e.g., captions), as the model better leverages the provided context to guide the generation process. Furthermore, an increase in $\Delta_{\text{preferred}} - \Delta_{\text{less-preferred}} $ reflects greater confidence in distinguishing preferred images from less-preferred ones, which translates to better preference-based optimization. Ultimately, minimizing the DCPO loss not only ensures more precise noise prediction but also improves alignment and image generation performance, enabling the model to produce outputs that are both reliable and realistic.
> > > >
> > > > We kindly ask the reviewer to suggest specific experiments or proofs that would address the concerns.
> > > >
> > > > ---
> > > > [1] From r to Q∗: Your Language Model is Secretly a Q-Function: https://arxiv.org/abs/2404.12358

---

> > ### Author Response · Authors · 2024-12-01
> > **Formal Proofs on DCPO's Motivation and Methodology - Part 2**
> >
> > **Proof 2 - Replacing caption $ c $ with the specifically generated caption $ z^l $ for the less-preferred image $ x_0^l $ decreases $\Delta_{\text{less-preferred}}$.**
> >
> > To analyze how replacing $ \mathbf{c} $ with $ \mathbf{z}^l $, where $ \mathbf{c} \subset \mathbf{z}^l $ and  $\mathbf{z}^l \sim Q(\mathbf{z}^l | x^l, c)$, for the less-preferred image $ \mathbf{x}_0^l $ improves the optimization, we delve into how the loss function is affected by this substitution.
> >
> > The term relevant to the less-preferred image $ \mathbf{x}_t^l $ in the loss is:
> >
> > $$
> > \Delta_{\text{less-preferred}} = \| \boldsymbol{\epsilon}^l - \boldsymbol{\epsilon}_\theta(\mathbf{x}_t^l, t, \mathbf{c}) \|_2^2 - $$
> >
> > $$\| \boldsymbol{\epsilon}^l - \boldsymbol{\epsilon}_{\text{ref}}(\mathbf{x}_t^l, t, \mathbf{c}) \|_2^2.
> > $$
> >
> > Replacing $ \mathbf{c} $ with $ \mathbf{z}^l $ modifies the predicted noise term $ \epsilon_\theta (x_t^l, t, c) $ to $ \epsilon_\theta(\mathbf{x}_t^l, t, \mathbf{z}^l) $. Since $ \mathbf{z}^l $ better represents $ \mathbf{x}_t^l $, we have:
> >
> > $$
> > \| \boldsymbol{\epsilon}^l - \boldsymbol{\epsilon}_\theta(\mathbf{x}_t^l, t, \mathbf{z}^l) \|_2^2 < $$
> >
> >
> > $$ \| \boldsymbol{\epsilon}^l - \boldsymbol{\epsilon}_\theta(\mathbf{x}_t^l, t, \mathbf{c}) \|_2^2
> >  \ [Eq.1].
> > $$
> >
> > When $\|\epsilon^l - \epsilon_\theta(x_t^l, t, z^l) \|_2^2$ becomes smaller,
> >
> > the term $ \Delta_{\text{less-preferred}} $ decreases. This leads to $\Delta_{\text{preferred}} - \Delta_{\text{less-preferred}}$ becoming larger, which improves the soft-margin optimization in the loss function $ L(\theta) $ that we have shown in Proof 1.
> >
> > We further elaborate on why $[Eq.1]$ is true. In the context of mean squared error (MSE) minimization, the optimal predictor of $ \boldsymbol{\epsilon}^l $ given some information is the conditional expectation:
> >
> > When conditioned on $ (\mathbf{x}_t^l, t, c) $:
> >
> > $$
> > \boldsymbol{\epsilon}_\theta^\ast(\mathbf{x}_t^l, t, c) = \mathbb{E}\left[ \boldsymbol{\epsilon}^l \mid \mathbf{x}_t^l, t, c \right]
> > $$
> >
> > When conditioned on $ (\mathbf{x}_t^l, t, z^l) $:
> >
> > $$
> > \boldsymbol{\epsilon}_\theta^\ast(\mathbf{x}_t^l, t, z^l) = \mathbb{E}\left[ \boldsymbol{\epsilon}^l \mid \mathbf{x}_t^l, t, z^l \right]
> > $$
> >
> > The total variance of $ \boldsymbol{\epsilon}^l $ can be decomposed as by the Law of Total Variance (conditional variance formula) [1]:
> >
> > $$
> > \operatorname{Var}\left( \boldsymbol{\epsilon}^l \right) = \mathbb{E}\left[ \operatorname{Var}\left( \boldsymbol{\epsilon}^l \mid \mathbf{x}_t^l, t, c \right) \right] + \operatorname{Var}\left( \mathbb{E}\left[ \boldsymbol{\epsilon}^l \mid \mathbf{x}_t^l, t, c \right] \right)
> > $$
> >
> > Similarly, when conditioning on $ z^l $:
> >
> > $$
> > \operatorname{Var}\left( \boldsymbol{\epsilon}^l \right) = \mathbb{E}\left[ \operatorname{Var}\left( \boldsymbol{\epsilon}^l \mid \mathbf{x}_t^l, t, z^l \right) \right] + \operatorname{Var}\left( \mathbb{E}\left[ \boldsymbol{\epsilon}^l \mid \mathbf{x}_t^l, t, z^l \right] \right)
> > $$
> >
> > Since $ c \subset z^l $, the information provided by $ z^l $ is richer than that of $ c $. In probability theory, conditioning on more information does not increase the conditional variance:
> >
> > $$
> > \operatorname{Var}\left( \boldsymbol{\epsilon}^l \mid \mathbf{x}_t^l, t, z^l \right) \leq \operatorname{Var}\left( \boldsymbol{\epsilon}^l \mid \mathbf{x}_t^l, t, c \right) [Eq.2]
> > $$
> >
> > This inequality holds because conditioning on additional information ($ z^l $) can only reduce or leave unchanged the uncertainty (variance) about $ \boldsymbol{\epsilon}^l $.
> >
> > The expected squared error when using the optimal predictor is equal to the conditional variance:
> >
> > $$
> > \mathbb{E}\left[ \left\| \boldsymbol{\epsilon}^l - \boldsymbol{\epsilon}_\theta^\ast(\mathbf{x}_t^l, t, c) \right\|_2^2 \right] = \mathbb{E}\left[ \operatorname{Var}\left( \boldsymbol{\epsilon}^l \mid \mathbf{x}_t^l, t, c \right) \right]
> > $$
> >
> > Similarly,
> >
> > $$
> > \mathbb{E}\left[ \left\| \boldsymbol{\epsilon}^l - \boldsymbol{\epsilon}_\theta^\ast(\mathbf{x}_t^l, t, z^l) \right\|_2^2 \right] = \mathbb{E}\left[ \operatorname{Var}\left( \boldsymbol{\epsilon}^l \mid \mathbf{x}_t^l, t, z^l \right) \right]
> > $$
> >
> >
> > From $[Eq.2]$, we have:
> >
> > $$
> > \operatorname{Var}\left( \boldsymbol{\epsilon}^l \mid \mathbf{x}_t^l, t, z^l \right) \leq \operatorname{Var}\left( \boldsymbol{\epsilon}^l \mid \mathbf{x}_t^l, t, c \right)
> > $$
> >
> > Taking expectations on both sides:
> >
> > $$
> > \mathbb{E}\left[ \operatorname{Var}\left( \boldsymbol{\epsilon}^l \mid \mathbf{x}_t^l, t, z^l \right) \right] \leq \mathbb{E}\left[ \operatorname{Var}\left( \boldsymbol{\epsilon}^l \mid \mathbf{x}_t^l, t, c \right) \right]
> > $$
> >
> > Therefore,
> >
> > $$
> > \mathbb{E}\left[ \left\| \boldsymbol{\epsilon}^l - \boldsymbol{\epsilon}_\theta^\ast(\mathbf{x}_t^l, t, z^l) \right\|_2^2 \right] \leq$$
> >
> > $$ \mathbb{E}\left[ \left\| \boldsymbol{\epsilon}^l - \boldsymbol{\epsilon}_\theta^\ast(\mathbf{x}_t^l, t, c) \right\|_2^2 \right]
> > $$

---

> > ### Author Response · Authors · 2024-12-01
> > **Formal Proofs on DCPO's Motivation and Methodology - Part 3**
> >
> > **continue of Proof 2:**
> >
> > ---
> > Assuming that the neural network $ \boldsymbol{\epsilon}_\theta $ is capable of approximating the optimal predictor
> >
> > $ \boldsymbol{\epsilon}_\theta^\ast $, especially as training progresses and the model capacity is sufficient, we can write:
> >
> > $$
> > \left\| \boldsymbol{\epsilon}^l - \boldsymbol{\epsilon}_\theta(\mathbf{x}_t^l, t, z^l) \right\|_2^2 \approx $$
> >
> > $$\left\| \boldsymbol{\epsilon}^l - \boldsymbol{\epsilon}_\theta^\ast(\mathbf{x}_t^l, t, z^l) \right\|_2^2
> > $$
> >
> > Similarly for $ c $
> >
> > $$
> > \left\| \boldsymbol{\epsilon}^l - \boldsymbol{\epsilon}_\theta(\mathbf{x}_t^l, t, c) \right\|_2^2 \approx $$
> >
> > $$\left\| \boldsymbol{\epsilon}^l - \boldsymbol{\epsilon}_\theta^\ast(\mathbf{x}_t^l, t, c) \right\|_2^2 .
> > $$
> >
> > Therefore, the expected squared error satisfies:
> >
> > $$
> > \mathbb{E}\left[ \left\| \boldsymbol{\epsilon}^l - \boldsymbol{\epsilon}_\theta(\mathbf{x}_t^l, t, z^l) \right\|_2^2 \right] \leq $$
> >
> > $$\mathbb{E}\left[ \left\| \boldsymbol{\epsilon}^l - \boldsymbol{\epsilon}_\theta(\mathbf{x}_t^l, t, c) \right\|_2^2 \right]
> > $$
> >
> >
> > Since the term of $ \Delta_{\text{less-preferred}} $ in the loss function involves the difference of squared errors, using $ z^l $ instead of $ c $ for the less preferred sample results in a lower error term:
> >
> > $$
> > \Delta_{\text{less-preferred}}^{(z^l)} = $$
> >
> > $$\left\| \boldsymbol{\epsilon}^l - \boldsymbol{\epsilon}_\theta(\mathbf{x}_t^l, t, z^l) \right\|_2^2 - $$
> >
> > $$\left\| \boldsymbol{\epsilon}^l - \boldsymbol{\epsilon}_{\text{ref}}(\mathbf{x}_t^l, t, z^l) \right\|_2^2
> > $$
> >
> > Comparing with the original:
> >
> > $$
> > \Delta_{\text{less-preferred}}^{(c)} = $$
> >
> > $$\left\| \boldsymbol{\epsilon}^l - \boldsymbol{\epsilon}_\theta(\mathbf{x}_t^l, t, c) \right\|_2^2 - $$
> >
> > $$\left\| \boldsymbol{\epsilon}^l - \boldsymbol{\epsilon}_{\text{ref}}(\mathbf{x}_t^l, t, c) \right\|_2^2
> > $$
> >
> > Assuming the reference model $ \boldsymbol{\epsilon}_{\text{ref}} $
> >
> > Remains the same or also benefits similarly from the additional information in $ z^l $, the net effect is that the first term decreases more than the second term, leading to a reduced $ \Delta_{\text{less-preferred}} $.
> >
> >
> > [1] Law of Total Variance (conditional variance formula): Ross, S. M. (2014). Introduction to Probability Models (11th ed.). Academic Press.
> >
> > ---
> > **Proof 3 - Replacing caption $ c $ with the specifically generated caption $ z^w $ for the preferred image $ x_0^w $ increases $\Delta_{\text{preferred}}$.**
> >
> > To prove that replacing $ \mathbf{c} $ with $ \mathbf{z}^w  \sim Q(z^w|x^w, c) $, where $ \mathbf{c} \subset \mathbf{z}^w $, for $ \mathbf{x}_0^w $ also contributes to a better optimized loss $L(\theta)$, we examine how this particular substitution affects the loss function.
> >
> > We let
> >
> > $$
> > R_\theta(\mathbf{c}) = \| \boldsymbol{\epsilon}^w - \boldsymbol{\epsilon}_\theta(\mathbf{x}_t^w, t, \mathbf{c}) \|_2^2,
> > $$
> >
> > $$
> > R_{\text{ref}}(\mathbf{c}) = \| \boldsymbol{\epsilon}^w - \boldsymbol{\epsilon}_{\text{ref}}(\mathbf{x}_t^w, t, \mathbf{c}) \|_2^2.
> > $$
> >
> > The rate of decrease in $ R_\theta $ due to $ \mathbf{z}^w $ is proportional to the model's ability to exploit the additional conditioning. Since $ \boldsymbol{\epsilon}_\theta $ is learnable,
> >
> > it can more effectively leverage $ \mathbf{z}^w $ than $ \boldsymbol{\epsilon}_{\text{ref}} $, yielding:
> >
> > $$
> > \Delta R_\theta = R_\theta(\mathbf{c}) - R_\theta(\mathbf{z}^w) \gg \Delta R_{\text{ref}} = R_{\text{ref}}(\mathbf{c}) - R_{\text{ref}}(\mathbf{z}^w).
> > $$
> >
> > We further elaborate on why the learnable model's noise prediction residual ($ R_\theta $) decreases faster than the reference model's residual ($ R_{\text{ref}} $) when $ \mathbf{c} $ is replaced by $ \mathbf{z}^w $. The residuals for the learnable and reference models are defined as:
> >
> > $$
> > R_\theta(\mathbf{c}) = \| \boldsymbol{\epsilon}^w - \boldsymbol{\epsilon}_\theta(\mathbf{x}_t^w, t, \mathbf{c}) \|_2^2,
> > $$
> >
> > $$
> > R_{\text{ref}}(\mathbf{c}) = \| \boldsymbol{\epsilon}^w - \boldsymbol{\epsilon}_{\text{ref}}(\mathbf{x}_t^w, t, \mathbf{c}) \|_2^2.
> > $$
> >
> > When $ \mathbf{c} $ is replaced with $ \mathbf{z}^w $ (where $ \mathbf{c} \subset \mathbf{z}^w $), the residuals become:
> >
> > $$
> > R_\theta(\mathbf{z}^w) = \| \boldsymbol{\epsilon}^w - \boldsymbol{\epsilon}_\theta(\mathbf{x}_t^w, t, \mathbf{z}^w) \|_2^2,
> > $$
> >
> > $$
> > R_{\text{ref}}(\mathbf{z}^w) = \| \boldsymbol{\epsilon}^w - \boldsymbol{\epsilon}_{\text{ref}}(\mathbf{x}_t^w, t, \mathbf{z}^w) \|_2^2.
> > $$
> >
> > The rate of decrease for each residual is defined as:
> > $$
> > \Delta R_\theta = R_\theta(\mathbf{c}) - R_\theta(\mathbf{z}^w),
> > $$
> >
> > $$
> > \Delta R_{\text{ref}} = R_{\text{ref}}(\mathbf{c}) - R_{\text{ref}}(\mathbf{z}^w).
> > $$
> >
> > The quality of conditioning, $ Q(\mathbf{c}) $, represents how well the conditioning $ \mathbf{c} $ aligns with the true noise $ \boldsymbol{\epsilon}^w $. We assume that
> > $$
> > Q(\mathbf{z}^w) > Q(\mathbf{c}),
> > $$
> >
> > where the improvement in conditioning quality $ \Delta Q $ is defined as
> > $$
> > \Delta Q = Q(\mathbf{z}^w) - Q(\mathbf{c}).
> > $$

---

> ### Author Response · Authors · 2024-12-01
> **Formal Proofs on DCPO's Motivation and Methodology - Part 4**
>
> **continue of Proof 3:**
>
> ---
> The residual for $ R_\theta $ is proportional to the misalignment between $ Q(\mathbf{c}) $ and $ \boldsymbol{\epsilon}^w $:
>
> $$
>  R_\theta(\mathbf{c}) \propto \frac{1}{Q(\mathbf{c})}.
>  $$
>
> Replacing $ \mathbf{c} $ with $ \mathbf{z}^w $ (higher $ Q $) results in a larger proportional reduction:
>  $$
>  R_\theta(\mathbf{z}^w) \propto \frac{1}{Q(\mathbf{z}^w)} \quad \text{with} \quad \Delta R_\theta \propto \Delta Q.
>  $$
>
>
> The reference model's residual $ R_{\text{ref}} $ depends weakly on $ Q(\mathbf{c}) $, as it is fixed or less adaptable:
>
> $$
> R_{\text{ref}}(\mathbf{c}) \propto \frac{1}{Q_{\text{ref}}(\mathbf{c})},
> $$
>
> where $ Q_{\text{ref}}(\mathbf{c}) $ is less sensitive to changes in $ \mathbf{c} $.
>
> Thus, the proportional improvement in $ R_\theta $ due to $ \Delta Q $ is significantly larger than for $ R_{\text{ref}} $.
>
> The preferred difference term is:
> $$
> \Delta_{\text{preferred}} = R_\theta - R_{\text{ref}}.
> $$
>
> As $ R_\theta $ decreases significantly more than $ R_{\text{ref}} $, the gap $ R_\theta - R_{\text{ref}} $ becomes larger, increasing $ \Delta_{\text{preferred}} $:
> $$
> \Delta R_\theta \gg \Delta R_{\text{ref}} \implies \Delta_{\text{preferred}} \text{ increases.}
> $$
>
>
> The learnable model $ \boldsymbol{\epsilon}_\theta $ benefits more from the improved conditioning $ \mathbf{z}^w $
>
> because of its adaptability and training dynamics. This results in a larger reduction in $ R_\theta $ compared to $ R_{\text{ref}} $. Mathematically, the relative rate of decrease:
> $$
> \text{Relative Rate} = \frac{\Delta R_\theta}{\Delta R_{\text{ref}}} \gg 1,
> $$
>
> which ensures that $ \Delta_{\text{preferred}} $ also increases, hence improving the optimization process in $L(\theta)$ and helping the model distinguish predictions on preferred and less preferred image-captions more effectively.
>
> ---
>
> We hope our responses have addressed the follow-up concerns over our motivations and methodology raised by the reviewers. All of the formal proofs in these 4 parts will become part of the finalized version of our paper.

---

### Note · Authors · 2024-12-16

I have read and agree with the venue's withdrawal policy on behalf of myself and my co-authors.